# Special delEVery: Extracellular Vesicles as Promising Delivery Platform to the Brain

**DOI:** 10.3390/biomedicines9111734

**Published:** 2021-11-20

**Authors:** Marie J. Pauwels, Charysse Vandendriessche, Roosmarijn E. Vandenbroucke

**Affiliations:** 1VIB Center for Inflammation Research, 9052 Ghent, Belgium; marie.pauwels@irc.VIB-UGent.be (M.J.P.); charysse.vandendriessche@irc.VIB-UGent.be (C.V.); 2Department of Biomedical Molecular Biology, Ghent University, 9000 Ghent, Belgium

**Keywords:** extracellular vesicles, brain targeting, drug delivery, neurological disorders, brain barriers

## Abstract

The treatment of central nervous system (CNS) pathologies is severely hampered by the presence of tightly regulated CNS barriers that restrict drug delivery to the brain. An increasing amount of data suggests that extracellular vesicles (EVs), i.e., membrane derived vesicles that inherently protect and transfer biological cargoes between cells, naturally cross the CNS barriers. Moreover, EVs can be engineered with targeting ligands to obtain enriched tissue targeting and delivery capacities. In this review, we provide a detailed overview of the literature describing a natural and engineered CNS targeting and therapeutic efficiency of different cell type derived EVs. Hereby, we specifically focus on peripheral administration routes in a broad range of CNS diseases. Furthermore, we underline the potential of research aimed at elucidating the vesicular transport mechanisms across the different CNS barriers. Finally, we elaborate on the practical considerations towards the application of EVs as a brain drug delivery system.

## 1. Introduction

The treatment of a broad range of central nervous system (CNS) pathologies is still severely hampered by the presence of multiple brain barriers [1,2]. Interestingly, several reports have already shown promising results using extracellular vesicles (EVs) as therapeutic delivery system to the brain. Here, we provide an overview of the present literature on both unmodified (i.e., EVs with natural brain targeting capacity) and modified EVs (i.e., EVs engineered to achieve enriched CNS targeting) and their applications. The included studies report on the intrinsic brain targeting potential of specific EVs, their brain barrier crossing capacity and their therapeutic effect in a broad range of CNS pathologies upon intravenous (IV), intraperitoneal (IP), subcutaneous (SC) or intranasal (IN) delivery. Local (invasive) brain administration reports are not discussed as we aimed to focus on traceable, appealing and clinically translatable routes. Additionally, tables are included that contain more details on the study methodology and we indicated which read-outs for brain targeting (i.e., biodistribution, cargo delivery and/or a therapeutic read-out) were used. The latter are important considerations as the presence of a (labelled) EV in the brain does not necessarily mean that the EV will successfully deliver its content whereas a sole therapeutic read-out only indirectly suggests brain delivery. Of note, despite the description of specific EV subtypes in many of the cited papers we decided to consistently use the nomenclature “EV” based on the current lack of subtype-specific markers [3]. We also highlight the potential to target alternative barriers next to the mostly studied blood–brain barrier (BBB) such as the blood–cerebrospinal fluid (CSF) barrier. Finally, we expand on several practical considerations towards EVs as a therapeutic delivery platform to the brain.

## 2. Barriers in Brain Drug Delivery

Multiple barriers protect the CNS from harmful peripheral factors, such as toxic molecules and invading cells or pathogens. In the brain, three crucial barriers are present: the BBB, the arachnoid barrier and the blood–CSF barrier. They separate the peripheral blood from the brain parenchyma, the CSF in the subarachnoid space (SAS) and the CSF in the brain ventricles, respectively. Multiple detailed reviews on these barrier structures can be found elsewhere [4,5,6,7,8]. Next to the brain, the CNS also comprises the spinal cord and the retina where the blood–spinal cord barrier and the blood–retina barrier are located [4,5]. While collectively protecting the nervous tissue against deleterious insults, these barriers also impede the passage of drugs and biologicals to the brain [1,9]. More than 98% of all small molecules and almost all larger molecules cannot cross the BBB [9]. Therefore, achieving sufficient barrier penetration and brain drug delivery is one of the biggest challenges in the development of CNS disorder treatments [10]. To improve drug delivery across the brain barriers, the field of nanomedicine has advanced towards the development of drug delivery platforms such as synthetic nanoparticles [11,12,13]. However, by focusing on the beneficial features of the “natural” EV delivery system, improved carrier systems might be developed to further enhance therapeutic drug delivery to the brain [14].

### 2.1. The Blood–Brain Barrier

The BBB is established by unique cerebral endothelial cells, tightly interconnected via complex tight and adherens junctions that strictly seal the paracellular pathway of the cerebral capillaries [15]. These cells, together with closely associated perivascular astrocytic end feet, perivascular neurons and pericytes, complete the neurovascular unit (NVU) interface that regulates the BBB physiology and passage [15]. While water and small hydrophilic molecules can cross the barrier via paracellular transport, small lipophilic molecules can easily passively diffuse [16]. Crossing of essential polar molecules such as glucose, amino acids and nucleosides is, however, only possible via transcellular carrier-mediated influx transporters. Bidirectional transport of larger hydrophilic compounds, such as proteins and peptides, is mediated by endocytosis and receptor mediated transcytosis (RMT) transport processes [17]. While some drugs have been modified and appeared to be successfully transported via carrier-mediated transporters (e.g., levodopa, the gold standard for the treatment of Parkinson’s disease (PD)), RMT is considered to be the transport mechanism that is most likely to achieve successful delivery [18]. In view of nanoparticle mediated delivery, several transport mechanisms have been described. These include transport via induced localized permeabilization of the BBB, passage through the endothelial cells via RMT, endocytosis of the nanoparticle into the endothelial cytoplasm followed by exocytosis, or a combination of these mechanisms. In case of RMT, several receptors have already been targeted (e.g., transferrin, insulin and low-density lipoprotein (LDL) receptors), by modification of the nanoparticles using targeting peptides, proteins or antibodies [11]. Nevertheless, the study of BBB transport is also complicated by several factors such as differential expression of transporters in different brain regions [19] or differential expression of receptors (e.g., transferrin receptor) between human and rodents [20,21,22]. Moreover, even though it is believed that the permeability properties of the BBB are mainly controlled by the endothelial cells [9], it is important to acknowledge that cell communication with other cell types (e.g., astrocytes, pericytes) contributes to a tighter barrier function. Therefore, these cell types constitute additional cell layers that need to be taken into account in drug BBB transport studies [23,24].

### 2.2. The Blood–Cerebrospinal Fluid Barrier

Next to targeting the mostly studied BBB, it was recently suggested that targeting other brain barriers such as the blood–CSF barrier might have added value in CNS drug delivery [25,26]. The blood–CSF barrier is located at the choroid plexus (CP) tissue, a cauliflower-like structure projecting into the four brain ventricles. The CP is responsible for the secretion of CSF and the regulation of numerous exchange processes to supply the brain with nutrients and hormones and to clear metabolites and toxic compounds from the brain. Furthermore, the CP takes part in neurohumoral brain modulation and neuroimmune interactions [27]. The CP is composed of cuboidal choroid plexus epithelial (CPE) cells that are apically facing the CSF, while at their basal side fenestrated blood vessels, that lack barrier properties, allow for free exchange of substances from and to the bloodstream. Therefore, in contrast to the BBB, the actual blood–CSF barrier function is provided by tight junctions at the apical side of the CPE cells. They firmly connect adjacent CPE cells to each other, thereby controlling paracellular transport. Hence, transcellular transport (passive diffusion, carrier mediated influx, transcytosis mediated transport) depends on specific and strictly regulated transport mechanisms. A few transport mechanisms that overlap with the BBB are described, including transcytosis via the transferrin receptor pathway, the insulin receptor pathway, LDL receptor pathways and LDL receptor-related protein (LRP) pathways (e.g., LRP1, LRP2 and LRP8). However, also some transporters unique for the blood–CSF barrier have been identified, such as folate (derivatives), indicating some alternative targeting and transport possibilities at this barrier to explore [25]. Moreover, the surface area of the blood–CSF barrier is greatly enhanced because of basolateral membrane infoldings and an extensive apical microvilli network at the blood and CSF side, respectively. As a consequence, in rat this surface was described to be half of the BBB surface area [28]. However, some structural aspects need to be kept in mind when considering the blood–CSF barrier for drug delivery. While crossing the BBB might lead to a wider brain distribution, targeting the blood–CSF barrier might provide primarily drug delivery to the CSF and subsequently the CSF-brain contact surfaces [25,29]. Consequently, this strategy might be more valid when the drug target is located in the ventricular, cisternal or subarachnoid spaces and surrounding areas. This might be the case for neuroinflammatory and infectious diseases, cerebral amyloid angiopathy, selected brain tumors, hydrocephalus or neurohumoral dysregulation [25]. Whether CSF delivered components might also reach deeper brain tissue penetration, is still a matter of dispute and needs to be further investigated [25,30,31]. In general, CSF distribution into brain tissue depends on diffusion only, which means the component concentration decreases logarithmically with distance [29]. However, the penetration also depends a lot on the exact component characteristics such as size, charge and lipophilicity [32]. Nevertheless, if a more continuous secretion or delivery could be achieved, extended diffusion and consequent deeper tissue penetration might be reached [27,33]. Overall, detailed studies are needed to characterize the transport mechanisms at the blood–CSF barrier. Furthermore, the influence of factors such as CSF clearance flow rate and other cell layers to overcome (e.g., the ependymal cell layer lining the brain ventricles, the pia mater and glia limitans layers at the brain surface and the perivascular spaces) need to be further investigated [25,26]. It was for example reported that the pial membrane does not seem to impede molecule entry from CSF to the brain, while the dense network of glial cells in the glia limitans may slow down diffusion or be a site of accumulation [34]. Interestingly, while vesicular transport seems limited in the cerebral endothelial cells [35], both clathrin-coated and non-clathrin-coated vesicles seem very abundant in the CP epithelium [36]. Indeed, Grapp et al. reported the transcytosis of folate-receptor alpha (FRα) positive EVs from the basolateral (i.e., blood facing) to the apical (i.e., brain facing) side of rat CPE cells and uptake of these EVs in the brain parenchyma [37]. The strong endocytic activity at the blood–CSF barrier is probably related to the extensive metabolic and synthetic functions of the CP [36]. Moreover, our research group identified the implication of CP derived EVs in blood to brain communication [38] and the importance of CP-mediated EV release in AD pathogenesis [39]. Furthermore, also signaling of CP derived EVs to neuronal stem cells (NSCs) in the brain neurogenic niche via miR-204 was reported [40]. Altogether, based on these observations the potential of targeting the CP transport mechanisms in a drug delivery context are worth further exploration.

### 2.3. Circumventing the Barriers

An alternative brain targeting strategy is the exploitation of methods to bypass the brain barriers, which can be achieved by, e.g., IN delivery. Indeed, IN administration of fluorescent tracers [41] and a variety of molecules (e.g., insulin, leptin) is indicative of successful CNS administration and therapeutic potential [42]. Mechanistically, brain delivery through this administration route occurs along the olfactory and trigeminal nerves which is believed to occur via intracellular (i.e., internalization by olfactory neurons followed by neuronal translocation to brain) and extracellular (i.e., crossing of olfactory epithelium followed by perineural transportation to brain) pathways [42,43]. Anyhow, the extracellular pathways are believed to be more important for brain delivery [44]. Administration of drugs via this route has several advantages, including the non-invasive nature allowing for self-administration by patients, a higher bioavailability and less side effects [45]. In contrast, disadvantages encompass the limited administration volume and the short retention time of drug absorption [46]. Other brain regions that harbor fenestrated capillaries and therefore lack brain barrier properties are the circumventricular organs (CVOs), located around the third and fourth brain ventricles [47]. However, brain delivery of compounds from the CVOs is limited due to the utilized mechanism of brain diffusion, as discussed in Section 2.2, and the presence of the tanycytic barrier lining the CVOs [48]. Furthermore, the small surface area of the CVOs in comparison with the brain blood vessel network makes them less suitable for brain drug delivery [36].

## 3. Extracellular Vesicles as Drug Delivery Vehicle

### 3.1. Extracellular Vesicles

EVs are membrane derived vesicles, secreted by a broad range of cell types. They are natural carriers of wide-ranging biological cargo including proteins, lipids, DNA, RNA and other bioactive molecules, particularly tuned by the type and physiological state of the parental cell [49,50]. While initially considered to be merely garbage bags [51,52], it has become clear that these vesicles have an important role in cell–cell communication between adjacent and distant cells [53,54]. After secretion, EVs can travel via body fluids such as blood, CSF and saliva to reach their specific target cells. Here, EVs elicit functional responses via interaction with surface receptors followed by signaling activation, or through cargo transfer after internalization or fusion with the recipient cell [50,54]. Consequently, EVs have been identified as important mediators in a broad range of biological and pathological processes. This has not only led to the recognition of EVs as a novel therapeutic target or biomarker source in many diseases, but also encouraged their exploration as new, natural based drug delivery system [54,55,56,57,58,59].

### 3.2. Extracellular Vesicles as Brain Drug Delivery System

Several EV characteristics are particularly appealing for their application as drug delivery vehicle in a broad range of CNS diseases [60,61]. They encompass factors such as endogenous biocompatibility, natural biological cargo protection, low immunogenicity and an ability to cross biological barriers [60,61], all factors that are still limiting the success of synthetic delivery systems [14]. However, the exact barrier crossing mechanisms whereby EVs can cross the brain barriers still need to be fully elucidated. A handful of research papers have reported on the involvement of specific interaction molecules, such as lymphocyte function-associated antigen-1 (LFA-1) on EVs interacting with intercellular adhesion molecule (ICAM)-1/C-type lectin receptor on endothelial cells [62], transferrin-transferrin receptor interactions [63], cluster of differentiation (CD)46 as major EV uptake receptor [64] or 6-mannose-receptor involvement for one specific EV type [65]. Moreover, some specific transport mechanisms including clathrin-dependent endocytosis [62,66], and caveolin dependent routes [62], RMT [63,65,67] and adsorptive transcytosis [65] have been identified (see Section 3.3.1, Table 1). However, to further elucidate the unique EV barrier shuttling properties, it is of utmost importance to identify both the involved receptors at the brain barrier cells and the EV–brain barrier interacting factors. Since brain barrier crossing capacity was reported for different EV types with a very varied composition, it seems a whole plethora of surface molecules might be involved [68]. For example, a whole range of integrins and tetraspanins might play an important role [60]. Importantly, this composition not only depends on the cell source and state, but even varies between EV subpopulations derived from the same cell source [69,70]. Optimized proteomic strategies are currently used to characterize the EV proteome, EV surface-proteome, protein topology, post-translational modifications and even interaction partners [71,72]. Advances in these fields will help us picture the molecular fingerprint of EV types and understand their functioning. Moreover, also other EV characteristics might contribute to the brain barrier crossing capacity such as size, zeta potential, lipid composition, protein corona and glycosylation [11,60]. More insight in how EVs are able to cross brain barriers could also be gained by the investigation of endogenous EV processes, such as their impact on brain barrier permeability [73]. Indeed, it has been described that EVs can both increase [74,75] or decrease brain barrier permeability [74]. Furthermore, also a certain degree of EV organotropism (i.e., the homing of specific EVs to their cellular or tissue of origin) has been described for different EV types. For example, tumor cell-derived EVs home to their cells of origin [67,76,77,78,79] or tumor tissue after systemic injection [77,80,81,82]. In addition, other EV types might display inherent targeting or tropism to a specific tissue or cell type [67,83,84,85,86].

Next to their natural cargo transfer capacity, EVs can also be loaded with a broad range of cargo molecules [58]. The currently applied loading methods can be divided into endogenous (i.e., pre-isolation) and exogenous (i.e., post-isolation) loading. In case of endogenous loading, the cargo is sorted into the EVs by the producer cell during biogenesis, for example after transfection of the producer cells. Interestingly, Silva et al. recently described a workflow to accurately quantify the efficiency of different EV-sorting proteins in EV cargo loading, at a single vesicle and single molecule level. They describe tetraspanin 14, CD63 and CD63/CD81 fused to the PDGFRβ transmembrane domain as the most efficient EV sorting proteins, suggesting increased potential in EV engineering [87]. This endogenous loading can also be applied for other cargo types, such as RNA molecules [56], or to achieve increased EV packaging [88]. In case of exogenous loading methods, loading of purified EVs is achieved via incorporation of the cargo onto or into the vesicles using methods such as co-incubation [89], electroporation [90] or by permeabilization via sonication, saponin, freeze–thaw and extrusion [91]. For a recent review and update on EV loading techniques we refer to [92]. Importantly, exogenous methods have been reported to have variable degrees of success. Indeed, aggregation of EVs or cargo was detected, which induced altered physiochemical and morphological characteristics [93]. This highlights the importance of EV characterization after application of the loading techniques, although it is often difficult to determine whether the cargo is loaded into, onto or just co-isolated with EVs [55]. While a lot of new loading mechanisms are being developed and efforts are being made to increase EV loading efficiency for many different cargo types, there is an increasing need for a standardized reporting frame to enhance the reproducibility and increase the chance of translational outcomes [94]. Up until now, a successful therapeutic read-out of loaded EVs has been described for several CNS diseases. Cargo types include small molecules (e.g., curcumin, paclitaxel, doxorubicin), nucleic acids (e.g., hsiRNAs, miRNAs, siRNA, circRNAs) and proteins (e.g., catalase, neprilysin). An overview of the different cargoes and the respective applied loading methods that have been explored in the treatment of brain/CNS diseases using unmodified or modified EVs can be found in Table 1 and Table 2.

Furthermore, EVs can also be surface engineered to achieve a higher or enriched brain targeting capacity. One way to achieve EV surface targeting ligand display is source cell transfection with engineered vectors, containing the gene of well-characterized and highly expressed EV membrane proteins, such as the lysosomal-associated membrane protein 2B (Lamp2b). The targeting moiety of interest will be expressed on the EVs via fusion to this protein. However, the downsides of cell engineering are the complexity, the higher production cost and the inapplicability of this technique to readily isolated EVs or modification of body fluid derived EVs. Therefore, direct biochemical functionalization via covalent conjugation, hydrophobic insertion or membrane permeabilization can be applied as alternative strategy [95]. These chemical modification strategies are less time consuming, easier and more efficient for large scale EV productions. For a more extensive overview on different and more recently developed EV surface modification strategies we refer to recently published reviews [95,96].

**Table 1 biomedicines-09-01734-t001:** Overview unmodified EVs applied for brain delivery. This table summarizes the experimental set ups and read-outs of the different studied EV types in brain delivery or brain disease EV treatment studies. Furthermore, information about the EV characterization based on the Minimal Information for Studies of EVs (MISEV) guidelines is included [3]. The “+” symbol in the protein marker column indicates the assessment of typical EV markers whereas the “-” symbol indicates the assessment of non-EV markers. Abbreviations: EV: extracellular vesicle; MoA: mode of action; ND: not determined; IV: intravenous; IP: intraperitoneal; IN: intranasal; UC: ultracentrifugation; SEC: size exclusion chromatography; TEM: transmission electron microscopy; AFM: atomic force microscopy; cpm: counts per minute; NTA: nanoparticle tracking analysis; DLS: dynamic light scattering; TRPS: tunable resistive pulse sensing; MRI: magnetic resonance imaging; CNP: cellular nanoporation; IVIS: in vivo imaging system; SRM: label-free selective reaction monitoring; Dil: 1,1′-Dioctadecyl-3,3,3′,3′-Tetramethylindocarbocyanine; DiR: 1,1′-dioctadecyl-3,3,3′,3′-tetramethylindotricarbocyanine iodide; CM: conditioned medium; PEG: Polyethylene glycol; PEI: polyethyleneimine; LDLR: Low density lipoprotein receptor; GBM: glioblastoma; DC: dendritic cell; NSC: neuronal stem cells; MSC: mesenchymal stem cells; ESC: embryonic stem cells; LPS: lipopolysaccharide; WGA: wheatgerm agglutinin; 6-OHDA: 6-hydroxydopamine; α-syn: alpha-synuclein; PFF: preformed fibril; PD: Parkinson’s disease; MCAO: middle cerebral artery occlusion; OGD: Oxygen-glucose deprivation; SCI: Spinal cord injury; HNSCC: head and neck squamous cell carcinoma; HPV: Human papilloma virus.

Cell Type	EV Source	Isolation Technique	EV Characterization			Experimental Set Ups	In Vivo EV Dose	Administration Route	MoA	Label or Loaded Cargo (Method)	Proof Brain /CNS Localization	Ref
			Morphology	Protein Markers	EV Size/Charge							
Brain cells	bEnd.3 endothelial cells (mouse)	Total Exosome RNA and Protein Isolation Kit	SEM	+CD9 +CD63 +CD81	<150 nm (SEM, Nano C nanosizing system–Beckman Coulter)	In vitro: bEnd.3 cells In vivo: Brain cancer zebrafish embryo’s	4 nl of 200 μg/mL EVs	IV (common cardial vein injection)	Active receptor-mediated endocytosis, not further specified	Rhodamine 123 label (incubation) Doxorubicin, Paclitaxel (incubation)	Yes, brain detection rhodamine labelled EVs (fluorescent confocal imaging) + effect tumor cells	Yang et al. [67]
		Total Exosome RNA and Protein Isolation Kit	ND (cfr. Yang et al. [67])	ND (cfr. Yang et al. [67])	ND (cfr. Yang et al. [67])	In vitro: bEnd.3 cells and astrocytes In vivo: Brain cancer zebrafish embryo’s	4 nl of 200 μg/mL EVs	IV (common cardial vein injection)	Possible involvement high expression CD63	Rhodamine 123 label (incubation) Anti-VEGF siRNA (EV transfection)	Yes, brain detection siRNA (fluorescent confocal imaging) + effect tumor cells	Yang et al. [86]
		ExoQuick-TC Exosome Precipitation Solution	ND	ND	ND	In vivo: Photothrombic stroke in type 2 diabetes mellitus mouse	3 × 10^10^ particles/mouse (qNano, iZon)	IV	ND	PKH26 label (incubation)	Yes, brain sections PKH26 labelled EVs (laser scanning confocal imaging) + functional effects endogenous EV miR-126	Venkat et al. [97]
	BV2 microglia (mouse)	Total Exosome Isolation Reagent and UC	TEM	+CD9 +CD63 +CD81	30–100 nm (TEM) 96 nm (NTA)	In vitro: Hippocampal neuron cells In vivo: Repeated mild traumatic brain injury mouse model	3 × 10^10^ particles in 200 μL/mouse, 35 days post-injury	IV	ND	PKH26 label (incubation) miR-124-3p (transfection source cells)	Yes, brain sections PKH26 labelled EVs (confocal imaging)	Ge et al. [98]
	BV2 microglia, M2 polarized (mouse)	UC	TEM	+CD9 +CD63 +TSG101	30–120 nm (TEM, NTA)	In vitro: Primary neural cells (OGD) In vivo: Transient MCAO mouse model	100 μg/dose/day/mouse, right after model induction, 3 consequent days	IV	ND	PKH26 label (incubation)	Yes, brain sections PKH26 labelled EVs (confocal imaging) + functional effects endogenous miR-124	Song et al. [99]
	Primary astrocytes (mouse)	UC	TEM	+CD9 +CD63 +ALIX	40–160 nm (DLS, Nanosizer)	In vitro: HT-22 neurons (OGD) In vivo: MCAO rat model	80 μg/2 mL, 1 h after ligation operation	IV	ND	Dil (only in vitro) (incubation)	Only functional effects	Pei et al. [100]
	Primary astrocytes, ischemic preconditioned (mouse)	UC	TEM	+CD9 +CD63 +ALIX +TSG101	50–150 nm (DLS, Zetasizer)	In vitro: Primary neural cells (OGD) In vivo: MCAO mouse model	100 μg EVs/day, 3 injections per day for total of 3 days, immediately after MCAO.	IV	ND	Dil (only in vitro) (incubation)	Dil labelled EV detection in brain mentioned + functional effects	Chen et al. [101]
	Primary pericytes (mouse)	UC	TEM	+CD9 +CD81	30–200 nm (NTA)	In vitro: Primary spinal cord endothelial cells. In vivo: SCI mouse model	20 μg EVs, 1 h after SCI	IV	ND	ND	Only functional effects (spinal cord)	Yuan et al. [102]
Cancer cells	MDA-MB-231 breast cancer cell line (brain seeking variant only) (human)	Ultracentrifugation and OptiPrep gradient UC	TEM	+CD9 +CD63 +ALIX -GM130	~158 nm (NTA)	In vitro: Static and microfluidic human brain endothelial cell models In vivo: Nu/Nu mice MDA-MB-231 cell injection model and zebrafish BBB model	3 μg EVs (3–4 × 10^9^ particles/100 μl) EVs, injected retro-orbitally, every 2 days for a total of 10 injections (mice) 5 nl of 400 μg/mL EV stock (zebrafish)	IV (retro-orbital) (Distribution studies and BBB integrity) Intracardiac (Transcytosis and BBB integrity)	Involvement of clathrin-dependent but not caveolin-dependent uptake for transcytosis	Gaussia luciferase/Palm TdTomato (Transduced source cells)	Yes, Mouse brain sections TdTomato labelled EV uptake Zebrafish EV transcytosis live imaging	Morad et al. [66]
	MDA-MB-231 breast cancer cell line (human)	Mini SEC	TEM	+CD9 +CD63 +ALIX +TSG101 -CANX -GRP94	76–130 nm (TRPS, Izon)	In vivo: CD-1 wild type mouse injections + LPS/WGA/M6P IP injection	1e6 1 × 10^6^ cpm of Iodine-125 radioactively labelled EVs	IV (left jugular vein)	Increased uptake after LPS stimulus possibly indicates involvement of selectins, cytokines, enhanced adsorptive transcytosis, insulin transport or disrupted barrier transport	Iodine-125 label (chloramine-T method)	Radioactively labelled EV measurement of whole brain and different brain regions	Banks et al. [65]
	SK-Mel-28 melanoma cell line (human)	ExoQuick-TC Exosome Precipitation Solution or MagCapture Exosome Isolation Kit PS	ND	+ALIX -GRP78 (SRM Analysis)	~105 nm (NTA)	In vitro: hCMEC/D3 cell model	NA	NA	Identification CD46 major receptor for uptake in human blood–brain barrier endothelial cells	PKH67 label (only in vitro) (incubation)	NA	Kuroda et al. [64]
	EL-4 lymphoblast cell line (mouse)	10,000 g centrifugation pellet + sucrose gradient centrifugation (post-loading)	ND	ND	ND	In vivo: 3 therapeutic models: -LPS brain inflammation -MOG-peptide induced EAE MS mouse model and -GL26-Luciferase brain tumor-bearing model mouse	10 μg EV protein/mouse	IN	ND	IRDye800 label DiR label PKH26 label (incubation) Curcumin or JSI-124 (incubation)	DiR labelled EV detection in brain (Odyssey laser scanning imager, Carestream Molecular Imaging system) Brain sections PKH26 labelled EVs (confocal imaging) Curcumin load detection in brain + functional effects	Zhuang et al. [103]
	SCCVII oral squamous cancer cells, (mouse)	Mini SEC	TEM	+CD9 +CD63 +ALIX +TSG101 -CANX -GRP94	76–130 nm (TRPS, Izon)	In vivo: CD-1 wild type mouse injections + LPS/WGA/M6P IP injection	1 × 10^6^ cpm of Iodine-125 radioactively labelled EVs	IV (left jugular vein)	Increased uptake after WGA stimulus suggests binding to brain endothelial cell glycoproteins containing sialic acid or *N*-acetyl-d-glucosamine + decreased uptake after LPS stimulus	Iodine-125 label (chloramine-T method)	Radioactively labelled EV measurement of whole brain and different brain regions	Banks et al. [65]
	MEL526 melanoma cell line (human)	Mini SEC	TEM	+CD9 +CD63 +ALIX +TSG101 -CANX -GRP94	76–130 nm (TRPS, Izon)	In vivo: CD-1 wild type mouse injections + LPS/WGA/M6P IP injection	1 × 10^6^ cpm of Iodine-125 radioactively labelled EVs	IV (left jugular vein)	No specific mechanism identified	Iodine-125 label (chloramine-T method)	Radioactively labelled EV measurement of whole brain and different brain regions	Banks et al. [65]
	PCI-30 Human HPV (-) HNSCC cell line (human)	Mini SEC	TEM	+CD9 +CD63 +ALIX +TSG101 -CANX -GRP94	76–130 nm (TRPS, Izon)	In vivo: CD-1 wild type mouse injections + LPS/WGA/M6P IP injection	1 × 10^6^ cpm of Iodine-125 radioactively labelled EVs	IV (left jugular vein)	Increased uptake after LPS stimulus possibly indicates involvement of selectins, cytokines, enhanced adsorptive transcytosis, insulin transport or disrupted barrier transport	Iodine-125 label (chloramine-T method)	Radioactively labelled EV measurement of whole brain and different brain regions	Banks et al. [65]
	SSC-90 Human HPV (+) HNSCC cell line (human)	Mini SEC	TEM	+CD9 +CD63 +ALIX +TSG101 -CANX -GRP94	76–130 nm (TRPS, Izon)	In vivo: CD-1 wild type mouse injections + LPS/WGA/M6P IP injection	1 × 10^6^ cpm of Iodine-125 radioactively labelled EVs	IV (left jugular vein)	No specific mechanism identified	Iodine-125 label (chloramine-T method)	Radioactively labelled EV measurement of whole brain and different brain regions	Banks et al. [65]
	Kasumi Leukemic cell line (human)	Mini SEC	TEM	+CD9 +CD63 +ALIX +TSG101 -CANX -GRP94	76–130 nm (TRPS, Izon)	In vivo: CD-1 wild type mouse injections + LPS/WGA/M6P IP injection	1 × 10^6^ cpm of Iodine-125 radioactively labelled EVs	IV (left jugular vein)	Increased uptake after WGA stimulus suggests binding to brain endothelial cell glycoproteins containing sialic acid or *N*-acetyl-d-glucosamine	Iodine-125 label (chloramine-T method)	Radioactively labelled EV measurement of whole brain and different brain regions	Banks et al. [65]
Stem cells	NSC (human)	Not described	TEM	+CD63 +CD81 (routinely detected–data not shown)	NSC < 200 nm (NTA)	In vitro: Differentiated neural cells In vivo: MCAO mouse model	3 doses (not specified) at 2, 14 and 48 h post TE_MCAO in young mice Or 6, 24 and 48 h post stroke (In aged mice)	IV	ND	Indium-111 Dil label (only in vitro (incubation)	Radioactively labelled EV detection 1 h post -TE-MCAO (SPECT) + functional effects	Webb et al. [104]
		UF	(cfr. Webb et al. [104])	+CD81 +NSC EV marker profile (MACSPlex exosome kit)	NSC < 200 nm (NTA)	In vitro: Human umbilical MSC. In vivo: MCAO porcine model	2.7 × 10^10^ particles/kg EVs in 50 mL, administered at 2, 14 and 24 h post-MCAO	IV (peripheral ear vein)	ND	Dil (only in vitro) (incubation)	Only functional effects	Webb et al. [105]
		UC	TEM	ND	~147 nm (NTA)	In vivo: 5xFAD AD mouse model	2.25 × 10^7^ particles, 1 or 2 injections	IV (retro-orbital)	ND	NA	Only functional effects	Apodaca et al. [106]
		UC	ND	ND	ND	In vivo: Wild type mice	6.70 × 10^6^ particles	IV, IN and hippocampal injection	ND	PKH26 (incubation)	PKH26 labelled EV detection in brain sections (confocal imaging)	Ioannides et al. [107]
	NSC (mouse)	PEG complexing and centrifugation	ND	ND	100–200 nm (NTA)	In vitro: Primary cortical astrocyte or neuronal cultures (OGD) In vivo: MCAO mouse model	10 μg of total EV protein, 2 h after transient MCAO	IV	ND	NA	Only functional effects	Sun X et al. [108]
	Urine stem cells (human)	DC/UC	TEM	+CD9 +ALIX +TSG101 -GM130	~74 nm (Flow nanoanalyzer)	In vitro: Neural stem (OGD) study In vivo: Rat MCAO stroke model	1 × 10^11^ particles, 1 injection, 4 h post-MCAO	IV	ND	DiR (only in vitro) (incubation)	DiR labelled EV biodistribution (IVIS Spectrum)	Ling et al. [109]
Blood cells	Raw 264.7 macrophage cell line (mouse)	UC and SEC (post labeling)	TEM	+ALIX +TSG101 +LAMP2B	~90 nm (NTA) ~130 nm (DLS) −18 mV (DLS)	In vitro: hCMEC/D3 model In vivo: Wild type mice	4 × 10^5^ cpm of Iodine-125-labelled EVs (65 μg or 3 × 10^11^ EVs per batch)	IV	LFA-1 (EV) with ICAM-1 and C-type lectin receptor on brain endothelial cells	Iodine-125-label (Chloramine-T method) CM-Dil label (only in vitro) BDNF (incubation)	Radioactive labelled EV delivery Radioactive labelled BDNF EV cargo brain delivery	Yuan et al. [110]
		UC	TEM and AFM	+ALIX +CD63 -CANX	~110 nm, after loading 117 nm (NTA) −4.5 mV, after loading −4.9 mV (DLS)	In vitro: hCMEC/D3 model In vivo: -SD rats for tissue distribution and bioavailability study -C57BL/6 mice model -okadiak injection AD mouse model	Curcumin-EVs at 0.4 mg/kg (rat) Curcumin-EVs at 20 μg curcumin load/dose. 1 injection/day for 7 days (mouse)	Rat: IV Mouse: IP	LFA-1 (EV) with ICAM-1 on brain endothelial cells	Fluorescent curcumin (source cell incubation)	Fluorescent EV cargo detected in brain sections (confocal imaging) and brain tissue (IVIS spectrum imaging)	Wang et al. [111]
		Gradient centrifugation + SEC (Sepharose 6 BCL)	TEM and AFM	+TSG101	100 nm, after loading 100–200 nm (NTA) 100 nm, after loading 100-200 nm (DLS)	In vitro: PC12 neuronal cells In vivo: 6-OHDA injection PD mouse model	2.4 × 10^10^ particles/mouse for biodistribution. 1.2 × 10^9^ particles, 10 times every other day for PD mice treatment.	IN	ND	Dil label (incubation) Catalase (incubation, freeze/thaw, extrusion + in vivo read-out: saponin and sonication)	DiR labelled EV brain sections (confocal imaging) + functional effects catalase loading	Haney et al. [91]
	J774A.1 macrophage cell line (mouse)	Mini SEC	TEM	+CD9 +CD63 +ALIX +TSG101 -CANX -GRP94	76–130 nm (TRPS, Izon)	In vivo: CD-1 wild type mouse injections + LPS/WGA/M6P IP injection	1 × 10^6^ cpm of Iodine-125 radioactively labelled EVs	IV (left jugular vein)	Involvement of 6-mannose-receptor + Increased uptake after WGA stimulus suggests binding to brain endothelial cell glycoproteins containing sialic acid or *N*-acetyl-d-glucosamine	Iodine-125 label (chloramine-T method)	Radioactively labelled EV measurement of whole brain and different brain regions	Banks et al. [65]
	Primary T cells (human)	Mini SEC	TEM	+CD9 +CD63 +ALIX +TSG101 -CANX -GRP94	76–130 nm (TRPS, Izon)	In vivo: CD-1 wild type mouse injections + LPS/WGA/M6P IP injection	1 × 10^6^ cpm of Iodine-125 radioactively labelled EVs	IV (left jugular vein)	Increased uptake after LPS stimulus possibly indicates involvement of selectins, cytokines, enhanced adsorptive transcytosis, insulin transport or disrupted barrier transport	Iodine-125 label (chloramine-T method)	Radioactively labelled EV measurement of whole brain and different brain regions	Banks et al. [65]
	Autologous DC (mouse)	Differential UC and sucrose gradient (post-labelling)	TEM	+ALIX +TSG101	~100 nm (NTA)	In vivo: wild type NMRI or C57BL/6 mice	1 × 10^10^ particles/g mouse	IV	ND	DiR label (incubation)	DiR labelled EV biodistribution (IVIS spectrum)	Wiklander et al. [85]
	Bloodserum derived (predominantly produced by reticulocytes)	UC	TEM	+CD9 +CD63 +CD81	40–200 nm (TRPS, Izon science)	In vitro: bEnd.3 and SH-SY5Y cell line In vivo: 6-OHDA injection PD mouse model	18 mg/kg (with 1 mg EVs = about 4.16 × 10^11^ blood particles)	IV	Transferrin–transferrin receptor interaction (on both EVs and brain endothelial cells)	PKH67/PKH26 (in vitro only) (incubation) DiD label (incubation) Dopamine (incubation)	DiD labelled EVs in brain sections (confocal imaging)	Qu et al. [63]
	Reticulocytes naieve or PD patients (human)	SEC	ND	+ALIX	~200 nm (NTA)	In vitro: Primary mouse brain endothelial cells, N9 microglia In vivo: CD-1 wild type mice + LPS stimulus	300,000 cpm of labelled EVs	IV (jugular vein)	Adsorptive transcytosis	-Iodine-125 label (chloramine-T method) -Dil label (incubation)	Radioactively labelled EV measurement of whole brain Dil labelled EVs were detected on brain slices–only in LPS condition (confocal microscopy)	Matsumoto et al. [112]
Other cell types	HEKT293 cells (human)	UC	ND	+CD9 +CD63 +CD81	~ 96 nm, with luciferase construct addition 80 nm (NTA)	In vitro: Brain microvascular endothelial cells (BMEC cell line)	NA	NA	Internalization via clathrin-dependent and caveolae-dependent routes	PKH67 and PKH26 label (in vitro only) (incubation) Lactadherin-Gaussia luciferase (Transduction source cells)	NA	Chen et al. [62]
	NIH-3T3 fibroblast cell line (mouse)	Mini SEC	TEM	+CD9 +CD63 +ALIX +TSG101 -CANX -GRP94	76–130 nm (TRPS, Izon)	In vivo: CD-1 wild type mouse injections + LPS/WGA/M6P IP injection	1 × 10^6^ cpm of Iodine-125 radioactively labelled EVs	IV (left jugular vein)	Inidcations involvement of mannose-6-phosphate receptor + Increased uptake after WGA stimulus suggests binding to brain endothelial cell glycoproteins containing sialic acid or *N*-acetyl-d-glucosamine	Iodine-125 label (chloramine-T method)	Radioactively labelled EV measurement of whole brain and different brain regions	Banks et al. [65]
	HaCaT keratinocyte cell line (human)	Mini SEC	TEM	+CD9 +CD63 +ALIX +TSG101 -CANX -GRP94	76–130 nm (TRPS, Izon)	In vivo: CD-1 wild type mouse injections + LPS/WGA/M6P IP injection	1 × 10^6^ cpm of Iodine-125 radioactively labelled EVs	IV (left jugular vein)	Increased uptake after WGA stimulus suggests binding to brain endothelial cell glycoproteins containing sialic acid or *N*-acetyl-d-glucosamine	Iodine-125 label (chloramine-T method)	Radioactively labelled EV measurement of whole brain and different brain regions	Banks et al. [65]

**Table 2 biomedicines-09-01734-t002:** Overview modified EVs applied for enriched brain delivery. This table summarizes the experimental set ups and read-outs of the different modified or targeted EVs in brain delivery or brain disease EV treatment studies. Furthermore, information about the EV characterization based on the Minimal Information for Studies of EVs (MISEV) guidelines is included [3]. The “+” symbol in the protein marker column indicates the assessment of typical EV markers whereas the “-” symbol indicates the assessment of non-EV markers. Abbreviations: EV: extracellular vesicle; MoA: mode of action; ND: not determined; IV: intravenous; IP: intraperitoneal; IN: intranasal; UC: ultracentrifugation; SEC: size exclusion chromatography; TEM: transmission electron microscopy; AFM: atomic force microscopy; cpm: counts per minute; NTA: nanoparticle tracking analysis; DLS: dynamic light scattering; TRPS: tunable resistive pulse sensing; MRI: magnetic resonance imaging; CNP: cellular nanoporation; IVIS: in vivo imaging system; Dil: 1,1′-Dioctadecyl-3,3,3′,3′-Tetramethylindocarbocyanine; DiR: 1,1′-dioctadecyl-3,3,3′,3′-tetramethylindotricarbocyanine iodide; CM: conditioned medium; PEG: Polyethylene glycol; PEI: polyethyleneimine; LDLR: Low density lipoprotein receptor; GBM: glioblastoma; DC: dendritic cell; MSC: mesenchymal stem cells; ESC: embryonic stem cells; RVG: rabies virus glycoprotein; nAchRs: nicotinic acetylcholine receptors; α-syn: alpha-synuclein; PD: Parkinson’s disease; MCAO: middle cerebral artery occlusion; OGD: Oxygen-glucose deprivation.

Targeting Ligand	EV Source	Isolation Technique	EV Characterization			Experimental Set Ups	In Vivo EV Dose	Administration Route	MoA	Label or Loaded Cargo (Method)	Proof Brain/CNS Localization	Ref
			Morphology	Protein Markers	EV Size/Charge							
RVG	Immature DCs (mouse)	UC	TEM	+LAMP2B	80 nm (NTA, TEM)	In vitro: Neuro2a cells and C2C12 cells In vivo: Wild type mice	150 μg EVs with 150 μg siRNA cargo	IV	nAchRs targeting	-(Cy5 /cy3 labeled) GAPDH siRNA -BACE1 siRNA (electroporation) -RVG-lamp2b (transfection)	Cy3 GAPDH siRNA RVG EV cargo detection in coronal brain sections (confocal imaging) + Functional siRNA cargo delivery	Alvarez-Erviti et al. [90]
		UC	ND	ND	100 nm (NTA)	In vitro: Human SH-SY5Y cells expressing mouse α-syn-HA In vivo: Wild type and Tg13 PD mouse model	150 μg EVs with 150 μg siRNA cargo	IV	nAchRs targeting	Anti-α-syn siRNA (electroporation) -RVG-lamp2b (transfection)	Functional siRNA cargo delivery	Cooper et al. [113]
	Primary DCs (mouse)	UC	ND	ND	~90 nm (NTA)	In vitro: SH-SY5Y cells expressing GFP or S129D α-syn In vivo: C57BL6/C3H F1 mice α-syn PFF PD mouse model.	150 μg EV/dose loaded with 150 μg shRNA Set up 1: 1 injection, read-out after 45 days Set up 2: Second injection day 45, read-out day 90	IV	nAchRs targeting	-Anti-GFP shRNA -Anti- α-syn shRNA (electroporation) -RVG-lamp2b (transfection)	Functional effects cargo delivery	Izco et al. [114]
	HEK 293T cells (human)	Gradient centrifugation and UC (post-loading/post-labelling)	TEM	+ALIX	100 nm (TEM)	In vitro: Neuro2a cells, C2C12 control cells, primary neurons In vivo: α-syn PFF PD mouse model.	120 μg/mouse, weekly, during 4 weeks	IP	nAchRs targeting	-(Fluorescently labelled) F5R2 aptamer (PEI method) -CellVue Claret EV label (incubation) -RVG-lamp2b (transfection)	Labelled EV detection in cortex and midbrain on brain sections (confocal imaging) + Functional effects F5R2	Ren et al. [115]
		Exosome isolation kit (Invitrogen)	TEM	ND	90 nm (TEM) 85 nm (NTA)	In vitro: Neuro2 cells In vivo: Morphine injection addiction mouse model	200 μg EVs/ mouse optimal dose. Loaded with 0.14 pmol/μg siRNA 4 injections, every 2 days	IV	nAchRs targeting	-Opioid receptor Mu siRNA (transfection) -RVG-lamp2b (transfection)	Functional effects cargo delivery	Liu et al. [116]
		Differential UC and sucrose gradient (post-labelling)	TEM	+ALIX +TSG101	~100 nm (NTA)	In vivo: Wild type NMRI or C57BL/6 mice	1 × 10^10^ particles/g mouse	IV	nAchRs targeting	-DiR label (incubation) -RVG-lamp2b (transfection)	DiR labelled EV biodistribution (IVIS spectrum)	Wiklander et al. [85]
		UC	TEM	+CD63 +ALIX +TSG101 +LAMP2B -GM130	~100 nm (NTA)	In vitro: HEK293 cells In vivo: Photothrombosis stroke mouse model	200 μg EVs, 24 h post ischemia	IV	nAchRs targeting	Dil label (incubation) -Nerve growth factor protein and mRNA (transfection) -RVG-Lamp2b (transfection)	Dil labelled EV detection in brain and different cell types (fluorescence imaging)	Yang et al. [117]
		UC	TEM	+CD9 +CD63 +TGS101 +LAMP2B	~117 nm (NTA/DLS)	In vivo: Rodent and non-human primate ischemic stroke models	12 mg EVs/kg	IV	nAchRs targeting	-Dil label (incubation) -RVG-Lamp2b (transfection) -circSCMH1 (transfection)	-Dil labelled EV detection in brain and different cell types (fluorescence imaging) -qPCR detection circSCMH1 RNA in brain tissue -Functional effects cargo delivery	Yang et al. [118]
	HEK293T cells (human) (+partially validated in MSCs (human))	-For In vitro: CM without purification (debris spins only) -For NTA: SEC (Exo-Spin kit) -For in vivo: Donor cells implantation	ND	+CD9 +TSG101 +HSP90B (unpurified CM) (ELISA)	~100 nm (NTA)	In vitro: -nAchRs expressing HEK 293T cells -Neuro2A cell cultures + 6-OHDA In vivo: Striatal 6-OHDA injection PD mouse model.	NA	Subcutaneous donor cell implantation	nAchRs targeting	“EXOtic” delivery system: -EV booster genes at C-terminus CD63 -L7Ae at C-terminus CD63 -C/D box in 3′UTR NanoLuc/catalase mRNA -Cx43 mutant -lamp2b-RVG construct (all transfection) -RVG-lamp2b (transfection)	Functional effects cargo delivery	Kojima et al. [88]
Optimized RVG (higher degradation resistance)	HEK293FT cells (human)	UC	TEM	CD63	~130 nm (NTA)	In vitro: neuroblastoma cells	NA	NA	nAchRs targeting	PKH67 label (incubation) -RVG-lamp2b (transfection)	NA	Hung et al. [119]
T7-peptide or RVG	HEK293T cells (human)	ExoEasy Maxi Kit (Qiagen) + UC (post-loading)	SEM	ND	ND	In vitro: C6 neural cells In vivo: Intracranial tumor rat model	20 μg EVs loaded with 20 μg miRNA-21 anti-sense oligonucleotides	IV	T7 peptide: Transferrin receptor (on both BBB and glioblastoma tumor cells) RVG: nAchRs targeting	-Dil label (incubation) -(Fluorescently labelled) miRNA 21 anti-sense oligonucleotide (electroporation) -T7-peptide-Lamp2b (transfection) OR -RVG-Lamp2b (transfection)	Detection DiR labelled EVs on brain section (confocal imaging) and brain tissue (IVIS spectrum) + functional effects cargo delivery	Kim et al. [120]
4F-LDL peptide	Fibroblast cells L929 (mouse)	UC (max 14,000 g)	TEM	ND	200–300 nm (TEM), 300–325 nm (DLS), −10 mV zeta potential	In vitro: U87 cells and U87 glioma 3D spheroids In vivo: U87 glioma injection BALB/c nude mice	5 mg/kg mouse MTX, EV dose not specified	IV	LDLR overexpresssion on the BBB and GBM cell lines	-PKH26 label/DiR label (incubation) -4F-LDL peptide (EV membrane inserted via ApoA-I mimetic peptide 4F) -Surface KLA (pro-apoptotic) therapeutic glioblastoma peptide (4F EV insertion) -MTX (source cell loading)	DiR labelled EV detection in brain (IVIS Spectrum) + Functional effects MTX cargo	Ye et al. [121]
RGD-4C peptide	ReNcell VM, neural progenitor cell line (human)	UC	TEM	+ALIX +TSG101 -CANX	<200 nm (NTA)	In vitro: BV2 microglia In vivo: MCAO mouse model	100 μg total protein = 2.5–3.7 × 10^10^ particles per mouse. After 1 h of MCAO and 12 h of reperfusion Study therapeutic potential: 300 μg EVs, 12 h after reperfusion (in 200 μL).	IV	Targeting the ischemic lesion region (integrins activated endothelial cells)	-CSFE label -Cy5.5 label (click chemistry) -RGD-4C peptide (phosphatidylserine binding domains of lactadherin) (incubation) -TdTomato-labeled or Gluc-display (transduction)	Detection Cy5.5 labeled EVs in brain tissue (IVIS spectrum) Detection TdTomato labelled EVs in brain sections (confocal fluorescence imaging)	Tian et al. [122]
c(RGDyK) peptide	H9 ESC (human)	UC	TEM	+CD63 +ALIX +TSG101 -GM130	70 nm (unmodified and PTX loaded EVs (with peptide) 107 nm (no peptide) 125 nm (with peptide) (Flow nanoanalyzer)	In vitro: different cancer cell lines for EV uptake, glioblastoma cell lines (U87 U251 for anti-proliferative characteristics In vivo: Glioma mouse model	1 × 10^11^ particles/mL, 125 μL) every other day, during 2 weeks	IV (caudal vein)	Targeting αVβ3 integrin receptors, overexpressed on the surface of proliferating glioblastoma tumor endothelium	Dil label (incubation) -Paclitaxel (incubation) -c(RGD) peptide (chemical crosslining)	Detection Dil labeled EVs in brain tissue (IVIS spectrum)	Zhu et al. [123]
CDX peptide CREKA peptide	Embryonic fibroblasts (mouse) (MEF) Mouse bone marrow derived DCs	UC and OptiPrep density gradient UC	Cryo-EM	MEF: +CD9 +CD63 +TSG101	MEF: ~20–50 nm unmodified, ~70–110 nm (after CNP) (DLS)	In vitro: U87 and GL261 cells In vivo: U87 tumor model and GL261 tumor model	1 × 10^12^ particles, every 3 days, 10 days after tumor cell implantation	IV	CDX peptide: U87 tumor cell targeting CREKA peptide: GL261 tumor cell targeting	PKH26 label (incubation) Tumor targeting peptides, both in fusion with N-terminus CD47 transmembrane protein (CNP) mRNA (CNP)	Detection PKH26 labelled EV brain tumor uptake (IVIS Spectrum, Two-photon imaging) + functional effects	Yang et al. [124]
RGERPPR peptide (tumor targeting)	Raw 264.7 macrophage cell line (mouse)	UF, UC, and filtration sequentially	TEM	+CD63 +CD81	~120 nm (NTA) zetapotential -25 mV	In vitro: U251 and Bel-7404 cell target receptor cells In vivo: Glioma bearing BALB/c nude mice	200 μg (with 800 μg of both curcumin and SPION)	IV	Glioma NRP-1 receptor	-Dil label (incubation) -RGERPPR peptide (click chemistry) -Superparamagnetic iron oxide nanoparticles (SPION) and curcumin (electroporation)	Detection DiR labelled EVs in brain tissue +SPION detection (MRI) + functional effects	Jia et al. [125]

### 3.3. Extracellular Vesicle Brain Targeting

#### 3.3.1. Unmodified Extracellular Vesicles

Based on their natural barrier crossing capacity, EVs have been explored as a “natural” drug delivery platform to the brain. Importantly, the tissue distribution and clearance of EVs might be influenced by the route of administration [85]. IV administration has been the most popular administration route studied up until now and is still gaining interest [126]. In general, the majority of the IV injected EVs rapidly accumulates in the organs of the reticuloendothelial system (RES), such as liver and spleen, with a rather small proportion of the injected dose reaching the brain after systemic administration [85,127,128,129]. While more specific differences (e.g., the impact of EV size [128]) are currently being studied, further detailed study of the EV pharmacokinetics will be important for safe and efficient clinical application. Here, we discuss several reports providing evidence for the natural brain barrier crossing capacity or therapeutic efficiency of specific cell type derived EVs. They are summarized together with the specific experimental details in Table 1. Further investigation of the underlying mechanisms of the “natural” brain targeting capacity of these EVs will be interesting to unravel and apply in the optimization of engineered EV targeting as discussed below in Section 3.3.2 and Table 2. A schematic overview of the covered topics is provided in Figure 1.

Brain (cancer) cell-derived EVs

Based on the finding that parental cell markers are expressed on EVs and are described to be important in EV target tissue-specificity, Yang et al. hypothesized that EVs from brain cells most likely display brain cell specific targeting molecules enhancing delivery across the BBB [67]. Indeed, when studying the in vitro uptake of four different brain cell derived EV types by brain endothelial cells, a significantly higher uptake of brain endothelial bEnd.3 cell type EVs was observed. Based on 37 °C dependent transport, this was presumably mediated via active transport systems such as receptor mediated endocytosis. In vivo, bEnd.3 EVs widely distributed throughout the zebrafish brain upon cardinal vein injection. Furthermore, when loaded with doxorubicin the vesicles delivered their cargo to glioma tumors in a danio rerio cancer model [67]. The barrier crossing capacity could not be described for glioblastoma-astrocytoma (U-87 MG), neuroectodermal tumor (PFSK-1) or glioblastoma (A-172) cancer cell line derived EVs [67], even though, brain barrier crossing of glioblastoma EVs from brain to blood was observed before [130,131]. In a follow-up study, Yang et al. reported the ability of the bEnd.3 EV type to deliver vascular endothelial growth factor (VEFG) siRNA across the BBB into the brain of the U-87 cell glioblastoma-astrocytoma zebrafish model after cardinal vein injection. Consequently, successful inhibition of the cancer cell aggregation was achieved [86]. Additionally, a study by Venkat and colleagues described how IV injection of bEnd.3 derived EVs induced miR-126 dependent neurorestorative effects in mouse brain after stroke in type 2 Diabetes Mellitus mice [97]. Hence, these results indicate the endogenous restorative potential of this EV type. Recently, also the brain targeting capacity of IV administered microglial cell derived EVs was described. PKH26 labelled EVs co-localized with neuronal, microglial and astrocytic markers in the injured brain of a repetitive mild traumatic brain injury mouse model. Moreover, EVs derived from upregulated miR-124-3p expressing microglia cells reduced neurodegeneration via the Rela/ApoE signaling in hippocampal neurons and improved the cognitive outcome in this mouse model [98]. In a model of transient middle cerebral artery occlusion (MCAO), neuronal uptake of M2-microglia derived EVs was observed alongside attenuated ischemic brain injury and enhanced neuronal survival mediated via the miR-124 EV cargo [99]. Pei et al. described that primary astrocyte cell derived EVs ameliorate neuronal damage via autophagy regulation in vitro and in vivo in the MCAO rat stroke model, after IV administration [100]. Furthermore, in vitro and in vivo neuroprotective effects were also reported for EVs derived from ischemic-preconditioned astrocytes. After injection in the ischemic MCAO mouse model, a reduced infarct volume and attenuated neurobehavioral deficits were observed. The effects were reported to be mediated via the circular (circ)RNA SHOC2 EV cargo that inhibited neuronal apoptosis via autophagy regulation [101]. Lastly, pericyte cells possess stem cell characteristics in the NVU and, based on their close relationship, pericyte EVs can easily be taken up by endothelial cells [102]. Therefore, their therapeutic potential was recently studied in a spinal cord injury model. Indeed, beneficial EV effects included amongst others reduced pathological changes, improved motor function, blood flow and oxygen deficiency, protected blood spinal-cord barrier and reduced edema. Even though this report focused on the spinal cord and not the brain, the therapeutic potential of this specific EV type was clearly illustrated [102].

b.Cancer cell derived EVs

Interestingly, also the brain targeting capacity of brain metastasizing cancer cell line derived EVs was reported. Using static and dynamic in vitro BBB systems and the in vivo zebrafish BBB model, Morad et al. reported that the “brainseeking” variant of triple negative MDA-MB-231 breast cancer cell line derived EVs can breach an intact BBB [66]. Transcytosis was identified as the specific transport mechanism and the involvement of the recycling endocytic pathways in the transcellular transport was demonstrated using high spatiotemporal resolution microscopy. Furthermore, the authors report a tumor EV regulated increase in BBB EV transport efficiency via an induced decrease in brain endothelial Rab7 expression, thereby downregulating the endocytic pathway degradation route [66]. Furthermore, by studying the in vitro binding and internalization of EVs derived from the brain metastatic melanoma cell line (SK-Mel-28) in a human endothelial cell (hMEC/D3) BBB model system, the CD46 receptor was identified to be a major receptor for EV uptake in the endothelial cells. Reduction of CD46 expression by RNAi reduced the EV transport over the endothelial cells by twofold. The exact barrier crossing mechanism or transcytosis involvement was, however, not further studied in this work [64]. Finally, IN administration was reported to allow the brain delivery of mouse lymphoma cell line derived EVs [103]. EVs loaded with curcumin or JSI124 were able to reach microglial cells in naïve conditions and exert respectively reduced brain inflammation, disease progression and delayed tumor growth in lipopolysaccharide (LPS)-induced brain inflammation, MOG-induced auto-immune disease and the GL26 brain tumor mouse models [103].

c.Stem cell derived EVs

In general, one of the most studied unmodified EV types is undoubtedly the mesenchymal stem cell (MSC) derived EV [126]. MSCs and their secretions, including EVs, are known to have intrinsic neuroprotective, regenerative and anti-inflammatory properties [132,133]. They constitute a very potent EV source in several diseases, including neurodegenerative diseases, although the exact underlying mechanisms of their therapeutic effects are not yet fully understood [134,135,136,137,138]. In general, a smooth brain targeting and brain barrier crossing capacity of this specific unmodified EV type can be interpreted, even though an EV biodistribution read-out was not always included in these studies. Nonetheless, this conclusion is justified based on the extensive beneficial effects observed in the CNS tissue after IV or IN administration of MSC EVs in a broad range of diseases such as Alzheimer’s disease (AD), multiple sclerosis (MS), stroke, neuroinflammation, traumatic brain injury, spinal cord injury, perinatal brain injury and status epilepticus. Moreover, some studies specifically investigated the brain targeting potential of this EV type. For example, Betzer et al. used efficient gold nanoparticle labeling of MSC EVs and reported distribution data that indicated a superior brain accumulation after IN administration compared to IV administration of this EV type [139]. Furthermore, biodistribution studies of EVs labelled in the same manner in different brain pathology (i.e., AD, PD, stroke, autism) mouse models indicated specific targeting and accumulation in the pathological brain regions [140]. For a detailed overview of the large amount of research on the methodologies and therapeutic effects of MSC EVs in neurodegenerative diseases we refer to recently published extensive reviews [134,135,136,137,138]. While a lot of research is investigating the efficacy of MSC derived EVs in modulating neurological disorders, research on NSC derived EVs is rather scarce. Nevertheless, promising and robust data for this EV type has also been reported recently [141]. A study by Webb et al. in a thromboembolic stroke mouse model reported an even more pronounced therapeutic potential of the NCS EVs in direct comparison to MCS EVs. While single photon emission computed tomography (SPECT) studies indicated that the EVs reached the infarcted hemisphere one hour after IV injection and got cleared after 24 h (h), long-term therapeutic outcomes including overall neuroprotection, reduced tissue loss, reduced chronic inflammation, and improved motor and memory impairments were observed [104]. The same group also investigated the IV administration of human NSC EVs in a porcine MCAO model. Similarly, this resulted in neuroprotective effects, including white matter preservation and decreased lesion volume and brain swelling, in agreement with behavioral and mobility improvements [105]. Furthermore, also in a mouse MCAO model a significant reduction of post stroke brain infarct zones after IV injection of mouse NSC derived EVs was observed. Based on the performed mechanistic in vitro studies, neuroprotection was potentially achieved via astrocyte function preservation [108]. Moreover, in the 5xFAD transgenic AD mouse model neuroprotective effects and improved behavioral outcomes after IV administrations of primary mouse NSC derived EVs were described as well [106]. Interestingly, Zhang et al. reported that interferon-γ NSC pre-conditioning significantly alters the content and abilities of subsequently derived EVs. Local injection of the conditioned EVs in ischemic brain regions exerted more therapeutic benefits compared to unconditioned EVs [142]. Additionally, the potential of this EV type was further exemplified by a study that concluded an equal brain targeting and distribution after administration via different routes (i.e., IV, IN or brain injected) [107]. Finally, IV injection of human urinary stem cell derived EVs enhanced neurogenesis and reduced/alleviated neurological deficits in rats suffering from ischemic stroke. Here, the in vitro study of the underlying mechanisms suggested that neurogenesis might be partially attributed to the inhibition of histone deacetylase 6 (HDAC6) via miR-26a EV cargo transfer [109].

d.Blood cell derived EVs

Focusing on immune cell derived EVs, Yuan et al. reported the natural brain crossing capacity of in vitro Raw 264.7 murine macrophage derived EVs. Mechanistically, the crossing capacity was identified to be mediated via the interaction of LFA-1 on EVs with ICAM-1 on the endothelial cells, and additional interaction with carbohydrate binding C-type lectin receptors on brain microvessel endothelial cells. Moreover, a 3.1-fold faster and 5.8-fold increased EV brain accumulation was identified in the presence of brain inflammation, probably mediated via increased expression of ICAM-1 at the endothelial cells in this condition [110]. Using the same cell source and based on the same LFA-1 with ICAM-1 interaction, Wang et al. described the in vitro BBB (hCMEC cell transwell model) RMT mediated crossing capacity of curcumin loaded EVs. Furthermore, after IP injection in mice, a higher brain accumulation of the curcumin-EV formulation was detected compared to free curcumin. After seven days of consecutive IP curcumin-EV administration to the okadaic acid AD mouse model, a clear reduction in neuronal injury and ameliorated learning and memory capacity was observed. The effects were mediated via activation of the AKT/GSK-3β pathway, which inhibited Tau protein phosphorylation [111]. Based on efficient catalase delivery of BBB crossing macrophages to neuronal cells after systemic administration in a PD mouse model [143] and the identification of EV mediated catalase transfer in this process [144], macrophage EVs for brain drug delivery were also further explored by Haney et al. Especially since the authors expected macrophage derived EVs would have a higher capacity to avoid entrapment by mononuclear phagocytes, this EV type might display a prolonged circulation time and hence improved therapeutic efficacy [91]. Indeed, both after IV and IN EV administration, a wide distribution of a considerate amount of fluorescently labelled EVs was detected throughout the brain. Slightly higher signals were obtained in the IN set up. Moreover, EVs loaded with the catalase enzyme using different ex vivo methods induced significant neuroprotective effects upon repeated IN EV administration in the 6-hydroxydopamine (OHDA) injection PD mouse model [91]. Moreover, Raw 264.7 macrophage derived EVs loaded with curcumin could successfully inhibit reactive oxygen species (ROS) mediated neuronal apoptosis and BBB damage in a rat ischemic-reperfusion model [145]. Remarkably, macrophage derived EVs have also shown biological effects without loaded drug cargo, for example in axonal regeneration [146]. Furthermore, also autologous dendritic cell (DC) derived EVs were described to reach the brain. Wiklander et al. reported the detection of DiR fluorescently labelled EVs in the brain 24 h after IV administration [85]. EVs derived from DCs that were stimulated with low levels of IFNγ were able to reach the CNS after IN administration and to remyelinate normal and damaged brain [147]. Remyelination, presumably mediated by miR-219 enriched cargo, suggests the potential application of these EVs in MS [147,148]. DC derived EVs were also explored in EV engineering context (see Section 3.3.2), mostly based on their low immunogenicity profile [90]. Since DC EVs can present tumor peptides to cytotoxic T cells and regulate immune responses via the presence of major histocompatibility class I and II proteins and tetraspanin CD63 on their surface, they also have great potential in tumor treatment [149]. More general, Qu et al. reported that murine fresh serum derived EVs (predominantly released by reticulocytes) exhibit a natural brain targeting ability based on a transferrin-transferrin receptor interaction. Serum EVs loaded with dopamine achieved a more than 15-fold higher brain distribution compared to free dopamine, improved therapeutic outcomes and lowered systemic toxicity in the 6-OHDA PD mouse model [63]. Finally, the brain targeting capacity of serum derived EVs can be further illustrated by the observations that IV injection of serum EVs derived from PD mice [150] or hepatic ischemia reperfusion injury rats [151] in wild type animals introduce the exact same pathological hallmarks in the brain as in the diseased donor animals. Furthermore, also reticulocyte EVs derived from PD patients were described to cross the BBB after IV injection in mice whereby even higher EV amounts reached the brain after administration of an IP LPS stimulus [112].

e.Other cellular sources

Looking into the BBB crossing potential of other cell types, Chen et al. reported that lactadherin-luciferase modified HEKT293 cell derived EVs were able to cross in vitro brain microvascular endothelial cell (BMEC) monolayers, but solely under tumor necrosis factor (TNF) activated inflammatory conditions [62]. Mechanistically, confocal microscopy images in the absence and presence of various endocytosis inhibitors indicated mainly transcellular transport, based on endocytic internalization of the EVs and their co-localization with endosomal markers [62]. In a more recent study, Banks et al. examined and compared the brain barrier crossing ability of 10 different EV populations to look further into cell source specific EV characteristics and to define the influence of inflammation on the EV barrier crossing capacity [65]. Studied EVs were obtained from mouse, human, cancerous and non-cancerous cell lines to investigate their ability to cross the BBB. Each EV type was radioactively labelled and IV injected, followed by pharmacokinetics studies of the different EVs in blood and brain using multiple-time regression analysis of the EV brain/serum ratio signals over time. Interestingly, all EV types in this study were shown to cross the BBB, though at various rates (ranging over 10-fold) and involving various mechanisms, including transport via specific receptors or adsorptive transcytosis. The total BBB crossing capacity of the different EV populations varied from 58% to 92%. Additionally, the brain EV fluxes were suggested to be influenced by brain-to-blood efflux, for example as determined after intracerebroventricular (ICV) EV injections of HaCaT EVs. Interestingly, also differences in EV uptake by various brain regions were studied. Indeed, for example, primary T cells, MDA-MB-231, SCC-90 and SCCVII were significantly higher taken up in olfactory bulb compared to other EV types. Furthermore, the effect of several triggers on EV barrier passage was investigated. An enhanced uptake of six EV types and decreased uptake of one EV type was observed after LPS challenge, while wheatgerm agglutinin (WGA) modulated the transport of five EV populations and supposedly allowed passage via adsorptive transcytosis. Interestingly, mannose-6 phosphate inhibited the uptake of the J774A.1 EV type, illustrating the involvement of the mannose-6 phosphate receptor for the uptake of this specific EV type. Even though all of the studied and thus BBB crossing EV types expressed CD46 and ICAM-1, in general, the EV uptake rate could not be predicted by the presence of EV surface proteins such as CD46, AVβ6, AVβ3 and ICAM-1, nor by the species of the EV source, the cancer status or specific responses to LPS of WGA [65]. Consequently, these results suggest the more complex interactions and/or the involvement of additional factors. Importantly, next to BBB targeting and crossing, Grapp et al. studied the EV mediated 5-methyltetrahydrofolate (5-MTHF) transport over the blood–CSF barrier. Remarkably, they found that in an in vitro transwell system 5-MTHF was transported from the basal (i.e., blood facing) to the apical (i.e., CSF facing) side of rat Z310 CPE cells via FRα positive EVs. Interestingly, after in vivo ICV injection, Z310 cell derived FRα positive EVs were described to cross the ependymal cell layer, distributed throughout the brain parenchyma and were taken up by astrocytes and neurons. On the contrary, only low amounts of FRα negative EVs seemed to cross the ependymal cell layer and were only taken up by microglial cells [37].

Together, these results indicate that several unmodified EV types have a clear brain targeting, brain barrier crossing and therapeutic delivery potential. However, based on the comparison of different EV types, several factors such as EV characteristics (e.g., size, surface molecules, lipid composition, surface corona) and barrier interactions (e.g., uptake mechanisms, transport mechanisms, crossing rates, retention and disease state) have a great influence on the EV brain targeting and transport efficiency. More in depth, mechanistic characterization of the targeting and crossing mechanisms of different EV types is needed to improve our understanding of the underlaying biological mechanism. Subsequently, this knowledge can be applied to further improve CNS targeting via modified EVs.

#### 3.3.2. Modified Extracellular Vesicles

Rabies virus glycoprotein modified EVs

One way to achieve EV surface targeting ligand display is via engineered vectors containing the gene of well-characterized and highly expressed EV membrane proteins, such as Lamp2b, in fusion with the targeting moiety of interest. Until now, the most popular brain targeting ligand, the rabies virus glycoprotein (RVG) peptide, is described to selectively target neuronal cells, brain endothelial cells and promote the crossing of the BBB, supposedly via binding to the nicotinic acetylcholine receptors (nAchRs) [152]. The first evidence of successful brain targeting engineered EVs applying this approach was provided by Alvarez-Erviti et al. [90]. Here, immature DC derived EVs were transfected to express RVG on the EV surface in fusion with the Lamp2b membrane protein. Using electroporation, the engineered EVs were loaded with GAPDH or β-site amyloid precursor protein cleaving enzyme 1 (BACE1) targeting small siRNA, whereafter they were injected IV. The EVs were able to cross the BBB and induced a significant knock down of the target genes expression and BACE1 protein levels, the latter being a potential target for treatment of AD [90]. Using a similar approach, Cooper et al. obtained alpha synuclein (α-syn) siRNA loaded RVG modified EVs that induced a reduction in brain α-syn mRNA and protein levels after injection in both wild-type mice and in the S129D transgenic PD mouse model. Subsequently, this resulted in a significant reduction of intraneural protein aggregates, including in the dopaminergic neurons of the substantia nigra. Together, these observations indicate the potential of these modified EVs to delay and reverse α-syn pathological conditions [113]. Importantly, the efficiency of the applied siRNA electroporation EV loading strategy has been reported with variable degrees of success and altered EV characteristics [93]. Again, this indicates the importance of further loading strategy characterization and standardization. Hung and colleagues looked further into the Lamp2b modification mechanism that was first applied in the studies described above. More precisely, they observed degradation of EV surface peptides fused to Lamp2b, for example by endosomal proteases during biogenesis [119]. Consequently, they developed new targeting peptide-Lamp2b fusion proteins to include glycosylation motifs at different positions. The newly engineered EVs derived from HEK293 cells, transfected with the enhanced constructs, were shown to achieve higher EV surface ligand expression and an increased uptake in neuroblastoma cells, indicating an intact interaction of the enhanced surface construct with its target [119]. In a later study in 2018, an EV production boosted HEK 293T cell line was established, called EXOsomal Transfer into Cells (EXOtic), also incorporating the Lamp2b-RVG surface expression to target the vesicles to the brain [88]. Moreover, using a newly designed EV mRNA packaging method, nanoluc mRNA and later also catalase mRNA were successfully loaded in the EVs via CD63 protein interaction to study optimized EV mRNA delivery. In vitro and subsequent in vivo read-outs, established by subcutaneous implantation of the EV donor cells, confirmed functional EV mRNA delivery to the target cells. Moreover, also a therapeutic effect in 6-OHDA-induced PD and LPS-induced neuroinflammation models was observed, reflected by reduced expression levels of several inflammatory markers. Remarkably, even though the EVs were RVG modified, the authors did not observe a higher brain delivery of RVG modified EVs compared to unmodified EVs in this context [88]. Furthermore, also the potential of Lamp2b-RVG modified HEK293T cell derived EVs loaded with α-syn aggregation inhibition aptamers (F5R2) were studied. These EVs were able to establish reduced pathological α-syn aggregation in primary neuron cultures in vitro and upon repeated IP injections in vivo. In the latter condition, the RVG-aptamer EVs could be localized in the brain cortex and midbrain. Additionally, using the preformed fibril (PFF) neostriatum injection model of sporadic PD, a clear rescue of the mouse grip strength loss was observed. Importantly, low brain localization and a functional effect were also observed using the non-aptamer-RVG EVs, but not for the unmodified EVs [115]. In the same year, another study reported on therapeutic effects of IV injected RVG modified EVs in the same PFF-induced PD mouse model [114]. Here, primary DC derived EVs were loaded with anti-α-syn shRNA mini-circles, which harbor longer term effectiveness compared to anti-α-syn siRNAs. In vivo therapy involving two IV injections decreased the α-syn aggregation significantly in several brain regions. Moreover, this was combined with a reduced loss of dopaminergic neurons and improved clinical outcomes [114].

RVG modified EVs were also explored in other CNS disease settings, for example via downregulation of opioid receptor Mu (MOR), the primary target of analgesics which is involved in the primary reinforcing effects and addiction to opiates. Here, HEK293T cells were co-transfected to acquire Lamp2b-RVG surface expression and MOR siRNA loaded EVs. A successful decrease in MOR mRNA and protein levels was observed, both in vitro in Neuro2a cells and in vivo in mouse brain tissue 24 h after one IV injected siRNA EV dose. Consequently, also reduced addiction outcomes were observed in the animals [116]. Likewise, the RVG peptide was studied in an ischemic stroke context. HEK293T EVs, surface modified with Lamp2b-RVG and loaded with high-mobility group box 1 (HMGB1) siRNA were administered IV in the MCAO mouse model [153]. A reduction of TNF and apoptosis levels in the brain was reported, including an effective reduction of the infarct size [153]. In a later study, the same source cell and targeting strategy was applied with additional nerve growth factor (NGF) loading of the HEK EVs. Efficient EV delivery into the ischemic cortex consequently reduced microglial inflammation, promoting cell survival and increased doublecortin neuroblast marker expression [117]. Recently, these modified EVs were also transfected to carry circular SCMH1 RNA (circSCHMH1), followed by injection in mice and nonhuman primates. An improved post stroke functional recovery in both mice and monkeys was observed via an enhanced neuroplasticity and inhibited immune response induced by the circSCHMH1 cargo [118].

b.Other targeting strategies

While currently the RVG peptide is the most applied brain targeting ligand, also other brain targeting or brain tumor ligands have been explored. Kim et al. modified HEK293T EVs to express the transferrin receptor binding T7 peptide, also in fusion to the Lamp2b protein, to target the respective receptor which is present on both brain endothelial cells and glioblastoma tumors. Remarkably, a higher brain accumulation of the T7 peptide EVs compared to RVG peptide EVs was demonstrated after their IV injection in a Dil labelled EV biodistribution experiment. Moreover, the T7 modified EVs loaded with miRNA-21 anti-sense oligonucleotide induced a far more pronounced reduction in the tumor size in the orthotopic glioblastoma models compared to RVG modified and unmodified EVs [120].

Using a direct biochemical surface functionalization approach, successful BBB crossing and treatment of glioblastoma multiforme was described by Ye et al. [121]. Here, fibroblast EVs were modified to express the LDL peptide, targeting both brain and glioma low-density lipoprotein receptor, and the therapeutic (pro-apoptotic) Lys-Leu-Ala (KLA)-peptide. Mechanistically, this was achieved by fusing the targeting/therapeutic peptide to the ApoA-I mimetic peptide (L-4F), which associates with the phospholipid rich EV membrane after simple co-incubation. After methotrexate (MTX) loading of these modified EVs, both BBB crossing and drug accumulation at the glioma site were demonstrated in ex vivo and in vivo experiments [121].

Whilst a certain barrier crossing capacity of unmodified MSC derived EVs was described in Section 3.3.1, several studies have also explored enhanced brain delivery via modification of this EV type. For an extensive overview of these studies, we refer to [134,135,136,137,138]. Amongst others, RVG modification [154] and other ligands such as the cyclo(Arg-Gly-Asp-D-Tyr-Lys) peptide or c(RGDyK) peptide, which exhibits high affinity for integrin α_ν_β_3_ expressed at the activated microvessels in stroke, were explored [155]. Recently, the latter group also performed modification of neuronal progenitor cell derived EVs. After in vitro confirmation of the inherent anti-inflammatory properties of this EV type, also here RGD peptide surface modification was established via fusion to phosphatidylserine binding domains of lactadherin. After IV EV injection in the same mouse ischemic stroke model, a strong suppression of the inflammatory response was observed, possibly dependent on seven identified MAPK inflammatory pathway inhibiting miRNAs [122].

Apart from EV modifications based on the addition of brain targeting ligands, a few reports have described an increased brain and brain tumor targeting via addition of tumor targeting ligands to the EV surface. For example, Jia et al. reported how mouse macrophage derived EVs were surface modified using click chemistry to obtain more specific glioma targeting [125]. A smooth BBB crossing capacity of the EVs was observed, followed by specific RGERPPR peptide enhanced glioma targeting and penetration, which could significantly increase the antitumor effects mediated by the curcumin and superparamagnetic iron oxide nanoparticle EV load [125]. Furthermore, also glioblastoma targeted EV therapeutics were optimized using surface modification. Here, the intrinsic antitumor properties of embryonic stem cell derived EVs were further complemented with antiproliferative paclitaxel drug loading and glioblastoma targeting c(RGDyK) peptide surface modification. Indeed, after SC in vivo administration a significantly enhanced tumor targeting was observed compared to unmodified drug loaded EVs [123]. Finally, a restored tumor suppressor function, enhanced tumor growth inhibition and increased animal survival after IV injection of glioma targeted EVs was also reported. Glioma targeting was obtained via CDX peptide (FKESWREARGTRIERG) expression on mouse embryonic fibroblast EVs for U87 tumor cell targeting, and CREKA peptide expression on mouse bone marrow derived DC EVs for GL261 tumor cell targeting, via fusion to the N-terminal of the abundant CD47 transmembrane protein. Importantly, using a novel encapsulation strategy called cellular nanoporation, a significant increase in EV production and highly efficient mRNA transcript loading was achieved, allowing for successful transcriptional manipulation and hence promising therapeutic outcomes [124].

## 4. Practical Considerations towards EVs as a Therapeutic Delivery Platform to the Brain

Despite the promising results on both brain delivery and therapeutic efficiency of (un)modified EVs, several practical hurdles will need to be assessed before EVs can be implemented as a brain delivery platform in the clinic. As summarized in Figure 2, we will elaborate on a number of practical considerations defining the translatability of EVs towards clinical practice.

### 4.1. Storage Conditions

When aiming to use EVs as therapeutic moieties, their storage conditions are of utmost importance. Indeed, experimental outcomes should be interpretable independent of variability introduced by EV storage to enhance reproducibility and to allow drawing reliable conclusions. Furthermore, in future industrial contexts massive amounts of therapeutic EVs will be needed, whereby storage conditions should not affect EV functionality. However, up until now no common guidelines on optimal storage conditions are available. Only a limited number of studies have investigated the impact of storage on EV characteristics, obtaining highly variable results.

Indistinguishable Nanoparticle Tracking Analysis (NTA) measurements were achieved when EV samples were stored at 4 °C or −80 °C for short periods of time (i.e., 1, 2 and 4 weeks), indicating that these storage temperatures do not impact EV concentration [156]. In contrast, a gradual decrease in EV concentration in a time span of 28 days at both 4 °C, −20 °C and −80 °C was reported [157]. Similarly, a significantly decreased EV number upon storage at 4 °C for 28 days was found, although in this study the EV numbers remained constant when stored at −20 °C and −80 °C [158]. In agreement with the latter, unaltered EV levels and size were described after 45 days or 6 months of storage at −80 °C [159]. A decrease in median EV size with increasing storage periods up to 25 days, independent of storage temperature, was reported [160,161]. No difference in EV size was measured upon storage for 24 h at different temperatures [160,162]. Another study found a gradual widening of the size range of EVs stored at 4 °C, −20 °C and −80 °C along the course of 28 days [157]. Along those lines, 4 days of EV storage at 4 °C and even more so at −80 °C resulted in a significant increase in EV diameter [163]. Accordingly, transmission electron microscopy (TEM) images of EVs stored at −80 °C for 4 days revealed the presence of larger, aggregated EVs harboring multilamellar membranes [163]. Furthermore, decreasing EV concentrations with increasing freeze–thaw cycles indicate that next to storage time, other factors need to be kept in mind [157,162]. Freeze–thaw cycles, however, did not seem to affect EV size [161]. Storage at pH 4 or pH 10 for 24 h at 4 °C caused a reduction in both EV concentration and protein level in comparison with pH 7, whereas EV size was unaffected [162].

EV protein levels remained constant when stored at −80 °C for up to 28 days, whereas decreasing EV protein levels were observed when preserved at 4 °C or −20 °C [157]. Similarly, EV storage for 10 days at room temperature or 4 °C caused a clear reduction in protein content whereas this effect was less pronounced after storage at −70 °C [164]. Nonetheless, mass spectrometry (MS) analysis of EVs stored at 4 °C or −80 °C for 4 days revealed profound changes in EV protein content whereby a number of storage-labile proteins were lost or leaked into the storage supernatant [163]. Since the latter proteins were suggested to be present on the external surface of EVs [163], this may have an impact on the biodistribution of therapeutic EVs although the EV components that define biodistribution are commonly not yet defined. In vitro cellular uptake and in vivo peripheral biodistribution upon IV injection were only minorly affected when EVs were stored at −80 °C for up to 14 days and 28 days prior to labeling, respectively, in contrast to storage at 4 °C or −20 °C [157]. However, even storage at −80 °C severely affected the biodistribution pattern to the brain as evident by a markedly decreased fluorescent intensity in this region when the unlabeled EV storage time exceeded 7 days [157]. Of note, the influence of storage conditions on labeled EVs was investigated as well. Carboxyfluorescein diacetate succinimidyl ester (CFDA-SE) labeled EVs rapidly lost their fluorescent intensity when stored at 4 °C whereas the fluorescent signal remained stable up to the latest assessed timepoint (i.e., 1 month) when the EVs were stored at −80 °C [156].

Although these studies provide an insight on the effect of storage conditions on EV concentration, size, protein content, cellular uptake and biodistribution, they did not report on another indispensable factor for therapeutic applications: EV functionality. When using in vitro complement activation as a functional readout, EVs stored at 4 °C and −80 °C but not −20 °C or 37 °C for up to 25 days maintained their functionality [160]. Of interest in light of therapeutic applications, the functionality of EVs stored at 37 °C began to decrease after 4 days but was unaffected after 1 and 2 days of storage [160]. Another study made us of the antibacterial capacity of their EVs to investigate the influence of storage conditions on EV functionality. Here, storage at 4, 20 and −20 °C for 1, 7 and 28 days gradually affected the antibacterial activity of the EVs while a similar trend was seen for storage at −80 °C, although less pronounced [158]. The in vitro antitumor efficacy of siRNA loaded EVs remained constant when stored at −80 °C for up to 6 months, whereby incubation times for up to 48 h at room temperature or 4 °C after thawing could be maintained [159]. Of interest, storage of modified EVs (i.e., NGF@Exo^RVG^, being RVG expressing and NGF loaded EVs) for up to 3 months at −80 °C did not affect the functional delivery of the loaded cargo to recipient cells [117].

To enable storage, the isolated EVs were generally resuspended in PBS [156,157,159,160,161,162,163], as previously recommended [165]. Alternatively, resuspension in HBSS [158] or buffers included in commercial EV isolation kits [156] were applied. Although siliconized vessels were already advised as storage containers to prevent adherence of EVs to the container surface [165], only one of the discussed studies specifically mentioned LoBind Eppendorf tubes [156] whereas others indicated polypropylene microtubes [158], microcentrifuge tubes [164] and cryo-glass vials [159]. Several strategies to improve preservation of EVs are currently under investigation, including the addition of a cryoprotectant such as trehalose. Storage of EVs in trehalose-containing PBS prevented changes in EV number and size induced by consecutive freeze–thaw cycles and improved conservation of biological activity of EVs upon storage at −80 °C for one month [166]. Furthermore, trehalose can be used as a so-called lyoprotectant during lyophilization (i.e., freeze-drying) which provides an alternative to conventional EV storage in a buffer. Trehalose-lyoprotected, lyophilized EVs were shown to retain their biological activity [167], identifying this preservation method as an interesting approach to be explored further. For a more elaborate discussion on these alternative EV preservation methods, we refer the reader to other reviews [168,169,170].

In general, the data obtained by the different studies are heterogenous or even contradictory, making it difficult to formulate unanimous conclusions. Part of the differences are likely introduced by experimental variability in (1) the applied EV isolation techniques (e.g., different centrifugation-based protocols [156,157,158,159,160,161,163] or commercial protocols [156,162,164]) whereby the separated EVs were subjected to different extents of quality control, (2) methods to measure experimental readouts (e.g., EV size based on NTA techniques (NanoSight [157,159,161] and ZetaView [160,162]) or dynamic light scattering [163]), (3) EV sources and (4) sample handling prior to EV isolation. Indeed, the storage conditions of the source of EVs (i.e., biofluids or culture medium) are a factor that cannot be overlooked either. In case of biofluids, it is frequently infeasible to immediately proceed with EV isolation after sampling. This is particularly the case when making use of samples stored in biobanks. In addition, for culture medium often large amounts of material are needed to achieve the required EV concentration for therapeutic applications, implying cumulative culture medium collections. These arguments indicate that it might be favorable to store the EV liquid of origin for a substantial amount of time, whereafter EVs can be simultaneously separated from (pooled) samples. As reviewed elsewhere [169,171], the impact of storage conditions on several biofluids and cell culture medium prior to EV isolation is under investigation as well. Importantly, the International Society for Extracellular Vesicles (ISEV) set up several task forces for conditioned medium and biofluids (e.g., blood [172], urine [173], saliva and CSF), whereby general recommendations for sample storage and handling, amongst others, are proposed.

In conclusion, the available data suggest that EVs are either best used immediately after isolation, or alternatively after short-term storage at −80 °C. This implements that it is favorable to store the biofluid/culture medium instead of the isolated EVs when prolonged storage is needed. Nonetheless, comparative studies are warranted to confirm this notion. Of note, in particular studies investigating the impact of storage conditions on modified EVs are currently still scarce. In general, the available data on storage conditions underlines the need of meticulously reporting all experimental variables (e.g., storage time, storage container and number of freeze–thaw cycles) important for reproducibility, especially in the EV field which is highly susceptible to variability. In this sense, the ISEV task forces, the transparent reporting and centralizing knowledge in EV research (EV-TRACK) initiative and the minimal information for studies of EVs (MISEV) guidelines are indispensable to bring therapeutic EVs to the next level [3,174].

### 4.2. EV Isolation Techniques

Separating EVs from their source (i.e., biofluid, culture medium) is one of the most important steps when investigating EVs as a brain drug delivery system since successful EV isolation influences many downstream factors (e.g., biodistribution, EV loading). A wide variety of EV isolation techniques is available based on different EV properties ranging from density (e.g., centrifugation-based methods, density gradient centrifugation) to size (e.g., size exclusion chromatography, filtration) and biological composition (e.g., precipitation-based methods, affinity-based methods). Each method has its own specificity and efficiency whereby quality control measures are of utmost importance to verify the depletion of contaminants and the successful separation of EVs from a given source [3]. Nonetheless, without undermining the importance of meticulous characterization of the isolated EVs, it is known that different EV isolation methods yield EVs with varying morphology, size and proteomic profile [175]. This knowledge poses an inherent challenge in comparing results from published studies since they utilize a broad range of available EV isolation techniques (see Table 1 and Table 2). Indeed, different EV isolation techniques likely enrich for different EV subpopulations with a varying number of co-isolated contaminants, in turn affecting EV biodistribution and therapeutic capacity [128]. Furthermore, when aiming to use EVs as therapeutic moieties it will be key to balance two important goals: (1) isolation of a pure EV sample, which will probably require the combination of different EV isolation methods thereby enhancing processing time and implying loss of material in consecutive steps and (2) applying an isolation method that allows processing large sample volumes in a time-efficient manner, probably resulting in a less pure EV sample. Given the high number of EVs needed for brain targeting studies, the second approach may be preferable although the importance of EV purity to maintain the biological purpose will have to be determined. Similar to the EV storage conditions, currently no consensus has been reached about a preferred EV isolation technique when assessing therapeutic EVs. Nonetheless, we hypothesize that it is unlikely that a one-size-fits-all approach will become available but instead, isolation techniques may need to be tailored to the EV source and/or the intended purpose. Anyhow, an absolute prerequisite will be to retain the biological structure and function of EVs used for brain delivery approaches. In this sense, harsh ultracentrifugation-based techniques or chemical precipitation agents might be less suitable whereas gentle methods such as size exclusion chromatography seem more favorable. However, comparative studies are warranted to confirm these speculations.

### 4.3. Cellular Source and Immunogenic Potential

Potential immunogenic side effects are an important consideration for therapeutic applications, but also in the preclinical stage where frequently human-derived EVs are being tested in animal models. As is clear from Table 1 and Table 2, EVs derived from a variety of cellular sources are currently investigated for their brain targeting efficiency. From a practical point of view, the EV source should produce a large number of EVs under consistent, sustainable and easily to maintain culture conditions whereby large-scale cell production is feasible [176]. Furthermore, easy manipulation such as a high transfection efficiency is a major advantage for endogenous EV loading (see Section 3.2) whereas an elongated cellular lifespan circumvents variability introduced by replenishing cells [176]. Immortalized cell lines tick all these boxes, but often inherently raise safety concerns regarding immunogenicity and tumorigenicity. Nonetheless, (un)modified HEK293T cell derived EVs were shown to induce only a limited immune response and no observable toxicity upon repeated injection resulting in 10 injections over a time-span of 3 weeks in C57BL/6 mice [177]. A similar absence in toxicity and immunogenicity was observed with EVs isolated from suspension HEK Expi293F cells, both in vitro in a human hepatic cell line (i.e., HepG2 cells) and in vivo upon a single IV injection in BALB/c mice [178]. Analysis of a variety of toxicity parameters in an in vitro study confirmed the lack of significant toxicity of MSC-derived EVs, which for several readouts was not the case for bovine milk-derived EVs [179]. Furthermore, SC injections of EVs derived from E1-MYC cells (i.e., an immortalized MSC cell line) did not induce tumor formation in athymic nude mice, whereas IP injections did not affect tumor progression in an athymic nude mouse model of head and neck cancer xenografts either [180]. Similarly, no detectable immune responses were registered in C57BL/6 mice upon repeated IP injection (i.e., every 48 h for 3 weeks) with MSC-derived EVs containing siRNA targeting oncogenic Kras (called iExosomes) [159]. Of interest, these iExosomes were produced on a large-scale using a bioreactor culture and high-scale electroporation, enhancing clinical feasibility [159]. Collectively, these data support the safety of the investigated EVs for in vitro applications and in mouse models. Nonetheless, additional factors beyond the scope of these studies will need to be considered, including the effect of variations in the applied treatment regimen (e.g., administration route, EV dose, duration of the treatment). Of importance, repeated IV administration of increasing doses of EVs derived from suspension HEK Expi293F cells into macaques resulted in a much more rapid plasma clearance towards later administrations, which is suggestive of the development of an immune response over time [181]. As such, this observation highlights another factor to take into consideration, namely inter-species variation. Ultimately, EV safety will need to be tested in clinical settings as well. In this context, it is expected that ongoing clinical trials with EVs will shed light on this matter in the near future [59].

### 4.4. Alteration of Brain Barriers in Neurological Disorders

Aging, the major risk factor for several neurodegenerative disorders, is known to affect the morphology, integrity and functionality of the BBB and blood–CSF barrier [182]. Of importance, these changes appear to be even more pronounced in the context of neurodegenerative diseases including AD, PD and MS [183,184]. Hereby, compiling data not only suggests disruption of the BBB and blood–CSF barrier but also indicates alterations in the expression of several molecules (e.g., receptors, transporters) associated with functional changes [183,184,185]. Although the underlying mechanisms of the brain targeting capacity of peripherally administered EVs are still poorly understood, it can be hypothesized that these alterations may affect EV biodistribution. Intuitively, breakdown of the brain barriers rather seems to be an advantage for brain targeting. However, the accompanying pathological changes including an enlargement of perivascular spaces and a diminished CSF and interstitial fluid (ISF) flow, hinder therapeutic brain delivery and distribution [183]. Nonetheless, as evident from Table 1 and Table 2, several studies already indicated successful brain targeting and/or therapeutic effects in animal models for AD, PD, MS and stroke, amongst others. This suggests that the underlying EV brain targeting mechanisms either remain (partially) unaffected in these disease models, or that the occurring neuropathological changes do not impact brain targeting. In one of the previously discussed PD studies, the brain targeting ability of murine serum-derived EVs was shown to be mediated by a transferrin-transferrin receptor interaction [63]. Interestingly, several clinical trials investigating therapeutic approaches based on the transferrin receptor -the most widely studied target protein for RMT-based brain delivery- show promising results in Hunter syndrome [186], which is suggestive of sustained brain delivery in a disorder linked with neurological complications. Another indispensable factor in both ageing and neurological disorders is inflammation. Of interest, LPS-induced inflammation generally increased the uptake of IV injected EVs in several brain regions, independent of LPS-induced BBB disruption [65,110]. Similarly, increased EV crossing was observed in an in vitro BBB model upon TNF-induced inflammation [62]. Here, this effect was likely mediated by a TNF-mediated enhancement of both BBB permeability and transcytosis [62]. Together, these results are indicative of enhanced brain targeting in inflammatory conditions, which could be beneficial in neurological disorders characterized by brain inflammation. Although more research is warranted to expand the preclinical knowledge and translate the obtained findings to clinical settings, the currently available data are supportive of sustained EV brain targeting in pathological conditions.

### 4.5. Other Important Factors

In this review, we expanded on several practical considerations affecting the therapeutic applicability of EVs. However, there are obviously more factors to consider which we will briefly touch upon here. Conclusions about the biodistribution and brain entry of EVs are often based on tracing labelled EVs, whereby different methods can be applied for both tracing and labelling (Table 1 and Table 2). However, EV labelling strategies come with several pitfalls that complicate the formulation of robust experimental conclusions. Factors to keep in mind include unbound dye, aggregate formation, aspecific labeling and alteration of native EV properties, as reviewed elsewhere [187]. Furthermore, distribution of EVs to specific regions does not necessarily imply functional delivery of the EV cargo which will likely be a prerequisite for therapeutic applications. Specific strategies are required to deduce functional delivery [187], but they are often not implemented to substantiate conclusions in literature which adds a layer of complexity in judging whether a certain EV type has therapeutic potential. To illustrate this, in Figure 3 we provide an overview of the detection methods reported for the EV types summarized in Table 1 and Table 2. Indeed, while a vast number of studies have combined labelled EV/EV cargo tracing with proof of functional EV cargo delivery to support EV brain delivery, a large number of studies have based the targeting evidence on only one of these read-out approaches. Another factor we already briefly touched upon is the applied treatment regimen. The choice of EV dose, administration route and administration frequency influences EV biodistribution and therapeutic efficacy. Indeed, while IV administration has been the most explored administration route over the past decade [126], other systemic or local administration routes lead to different pharmacokinetics and pharmacodynamics, which might be more or less suitable in specific disease contexts. On top of that, the EV dose depends on the applied EV isolation and quantification method which enhances variation due to the wealth of available techniques. A meta-analysis of 64 preclinical studies revealed large variations in the applied EV dose and treatment regimen [188], highlighting another hurdle affecting comparison between studies and interpretation of the therapeutic capacity of the investigated EVs. Additionally, further research will be required to allow extrapolation of dosing and administration from preclinical models to humans [188]. A general notion that can be observed independent of the EV administration and dosing regimen is the limited EV circulation time whereby the majority of the EVs accumulate in the liver, lungs, kidneys and spleen [128]. While modifications to enhance brain targeting are discussed in Section 3.3.2, efforts are also being made to slow down EV clearance from the bloodstream. For example, EV modification with the “do not eat me” molecule CD47 [124,189,190] and PEGylation [191] have been shown to prolong EV circulation time. Several other candidates have been proposed, but further research will be needed to assess their effect on EV circulation [192]. Finally, to reach the ultimate goal of using EVs as therapeutic delivery vectors, some regulatory challenges will need to be addressed as well [126,193]. Although the road ahead towards EVs as clinical brain delivery vehicles is still long and filled with hurdles, the wealth of ongoing efforts (e.g., regarding EV production, characterization and standardization) in this rapidly evolving research field provides a hopeful mixture for the future. This notion is supported by the increasing number of ongoing clinical trials with therapeutic EVs, which may pave the way for several shared hurdles with EVs as drug delivery vehicles.

## 5. Conclusions

Throughout this manuscript, we discussed the current literature describing the potential of EVs as drug delivery vehicle in the treatment of several CNS diseases. It is clear that different EV types, with or without further modification or loading, have proven to reach the CNS and effectively elicit a therapeutic response upon peripheral administration. Nevertheless, even though certain EV uptake receptors and transport pathways have been identified, our knowledge of the EV CNS targeting mechanisms remains fairly limited. Therefore, further study into (1) contributing factors such as EV heterogeneity, membrane composition or other physicochemical aspects of specific EV types that display enhanced CNS targeting and (2) CNS barrier crossing mechanisms, will enhance our understanding of CNS barrier interactions and transport strategies. Moreover, by pursuing the combination of both EV tracing and proof of functional cargo delivery in further research, stronger evidence of the applicability of different (un)modified EV types should be gathered. Consequently, these insights will further improve the currently explored therapeutic EV delivery approaches to the brain and even specific targets within the brain. This, in combination with the extensive ongoing efforts towards more efficient and standardized EV production, well considered cell source choices (with or without inherent therapeutic features), detailed EV characterization, implementation of advanced study methods, improved targetability engineering, increased possibilities to load various drug cargos, determination of optimal administration routes, pathology implications and thorough safety assessment, all point towards the promising possibilities of EV based treatment of a broad variety of CNS diseases.

## Figures and Tables

**Figure 1 biomedicines-09-01734-f001:**
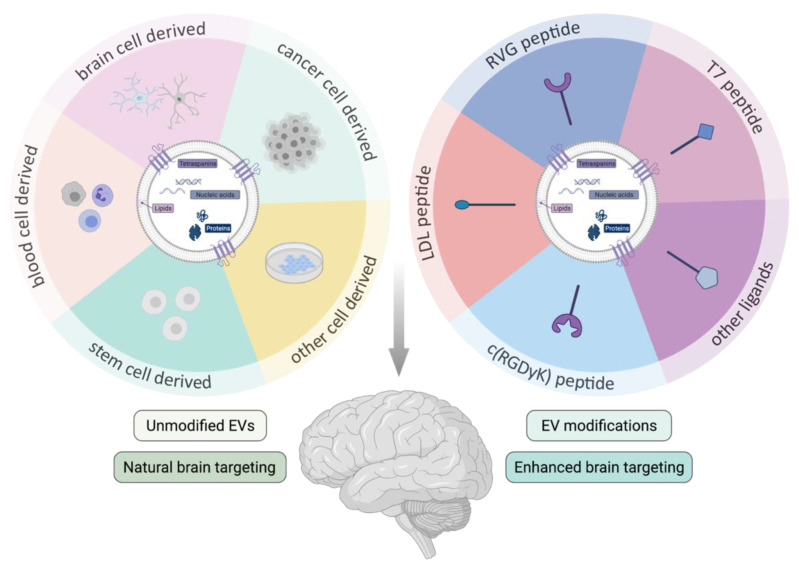
Schematic overview of the discussed literature on (un)modified extracellular vesicles (EVs) in this review. Unmodified EVs derived from various specific cell sources are described to possess a natural brain targeting capacity. Additionally, EVs can be engineered with diverse targeting ligands to obtain enriched brain targeting.

**Figure 2 biomedicines-09-01734-f002:**
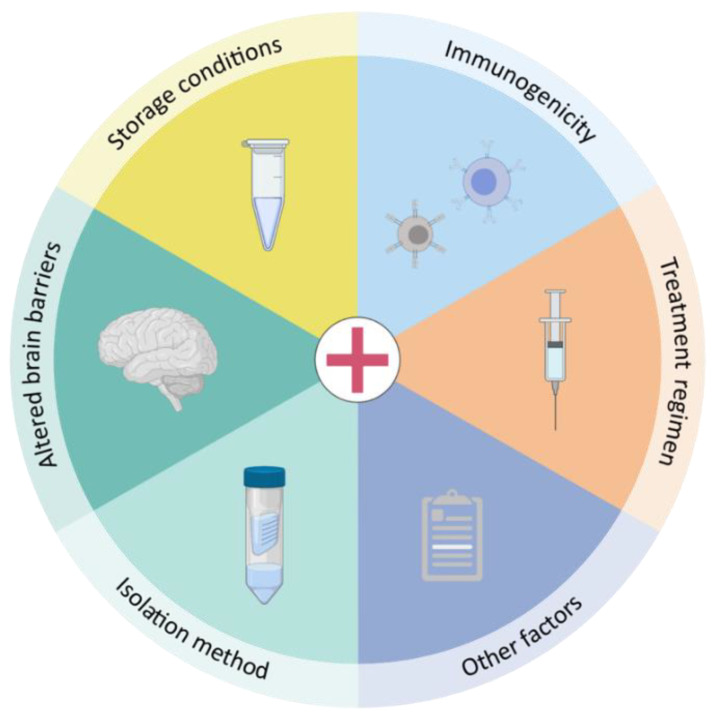
Schematic overview of the discussed practical considerations towards extracellular vesicles (EVs) as a therapeutic delivery platform to the brain.

**Figure 3 biomedicines-09-01734-f003:**
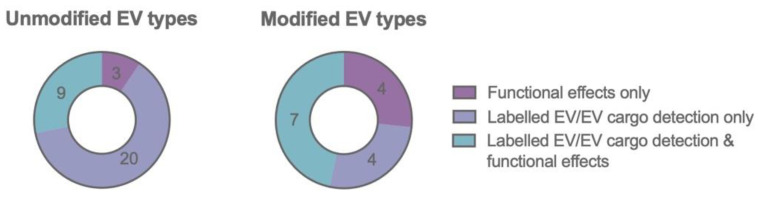
Overview of the brain targeting extracellular vesicle (EV) detection methods of the summarized literature (Table 1 and Table 2). Pursuing the combination of both EV tracing and proof of functional cargo delivery will provide stronger evidence of EV brain targeting capacities.

## Data Availability

Not applicable.

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
