# Peer review of "Special delEVery: Extracellular Vesicles as Promising Delivery Platform to the Brain"

_biomedicines, 2021, doi:10.3390/biomedicines9111734_

Round 1

Reviewer 1 Report

The review by Pauwels et al is informative and well written.

I have only minor comments:

1) It has been shown that patients derived EVs can be loaded with diagnostic agents. This strategy could support the early tumor detection and subsequent treatment (Villa et al. Theranostics 2021)

2) The usage of EVs is still far from the clinic. How realistic is the GMP production and the clinical translation (heterogeneicity, scale-up, EV-dose)

Author Response

The review by Pauwels et al is informative and well written.
I have only minor comments:
1. It has been shown that patients derived EVs can be loaded with diagnostic agents. This strategy
could support the early tumor detection and subsequent treatment (Villa et al. Theranostics
2021)
The suggested reference was added in revised manuscript line 220.
2. The usage of EVs is still far from the clinic. How realistic is the GMP production and the clinical
translation (heterogeneicity, scale-up, EV-dose)
We indeed agree with the reviewer that the currently existing procedures are still inadequate.
However, although we can’t predict the future, we believe that the wealth of ongoing efforts (e.g.
regarding EV production, characterization and standardization) in this rapidly evolving research field
provides a hopeful mixture for the future. This notion is supported by the increasing number of
ongoing clinical trials with therapeutic EVs, which may pave the way for several shared hurdles with
EVs as drug delivery vehicles. This was included in revised manuscript line 946:
“Although the road ahead towards EVs as clinical brain delivery vehicles is still long and filled with
hurdles, the wealth of ongoing efforts (e.g. regarding EV production, characterization and
standardization) in this rapidly evolving research field provides a hopeful mixture for the future. This
notion is supported by the increasing number of ongoing clinical trials with therapeutic EVs, which may
pave the way for several shared hurdles with EVs as drug delivery vehicles.”

Reviewer 2 Report

The review manuscript by Dr Pauwels and colleagues provides a comprehensive summary of the current state of EVs as a delivery system for brain diseases and discusses the unique potential of EVs to cross brain barriers as well as the challenges in the field. Although there are already plenty of reviews on EVs and similar reviews on EV-BBB delivery have been published elsewhere, this manuscript could be still an informative addition. Nevertheless, I have the following comments for the authors to address:

Major comments

  1. The title “Special delEVery…” looks odd and does not help to convey the message, suggest to change the word back “delivery”.
  2. It would be helpful to draw a diagram illustrating brain barriers, EV – brain delivery and potential cellular mechanisms in various brain diseases.
  3. Tables 1/2 provide overviews of brain delivery with native/modified EVs, since there is a lot of hype instead of facts in the EV-brain delivery field, it would be necessary for the authors to utilize the rich database in these two tables for a deeper analysis, e.g. it would be informative to have figures presenting how many studies provided solid data on EV distribution in the brain area and if the BBB remains intact in the specific animal models employed in those studies. A sub-analysis based on the delivery routes would be helpful too.
  4. EV storage is critical for their applications. While the authors spent 3 big paragraphs to identify the issues with all conventional storage approaches, unexpectedly, they only briefly mentioned the emerging effective EV preservation approaches in one sentence and referred to one review in 2018. In my opinion, the authors are expected to spend more effort in elaborating the emerging lyophilization approach to provide/propose the readers with “solutions” instead of repeating “problems” that have been repeated many times in other reviews. The authors are recommended to discuss two recent efforts (PMID: 27824088; PMID: 34371743) on EV/EV memetics lyophilization and the preservation of EV features both in vitro and in vivo at 4C or room temperature for long term storage.
  5. In section 4.2, it is indeed wise to provide a brief summary of “EV isolation techniques” instead of a much detailed discussion as this topic has been extensively reviewed elsewhere (ref 3, 95, 168, and a very recent comprehensive comparison on the pros & cons of current EV isolation methods by PMID: 34229852). However, the authors may elaborate the discussion by providing the audience certain guidance specifically for brain delivery based on the pros & cons of available EV isolation approaches: what types of approaches may be better than the others, in what circumstances?

Minor points

  1. English grammar checks are needed.
  2. Section title of 3.3 should be rephrased to make it concise.
  3. Subsection “3.3.2Brain (cancer) cell derived EVs” is not at the same level as 3.3.1, suggest to include it as part of subsection 3.3.1.

Author Response

1. The title “Special delEVery…” looks odd and does not help to convey the message, suggest to
change the word back “delivery”.
Although we appreciate the suggestion, we would like to keep the title of our review since we believe
this is an original way to spark the interest of readers. Furthermore, we are confident that the
reference towards “EVs” in “delEVery” will be clear to researchers in the EV field.
2. It would be helpful to draw a diagram illustrating brain barriers, EV – brain delivery and
potential cellular mechanisms in various brain diseases.
In our review we focused specifically on the emerging evidence of the brain targeting capacity of
different EV types. Since there are many concise and specialized reviews available on brain barriers,
we added clearer references to these specific reviews in revised manuscript line 52.
“Multiple detailed reviews on these barrier structures can be found elsewhere [1-5].”
Moreover, based on the currently available literature we concluded that rather few specific molecular
mechanisms of brain barrier crossing are known and fully described up until now, let alone the exact
and specific mechanisms employed by EVs. We therefore decided to include the reported mechanistic
details in tables 1 and 2, without making a separate diagram for these rather few (or incompletely
revealed) mechanisms. Therefore, we believe that the requested figure is out of the scope of this
review.
3. Tables 1/2 provide overviews of brain delivery with native/modified EVs, since there is a lot of
hype instead of facts in the EV-brain delivery field, it would be necessary for the authors to
utilize the rich database in these two tables for a deeper analysis, e.g. it would be informative
to have figures presenting how many studies provided solid data on EV distribution in the brain
area and if the BBB remains intact in the specific animal models employed in those studies. A
sub-analysis based on the delivery routes would be helpful too.
We agree with the comment of the reviewer that the facts on EV-brain delivery are often limited and
overstated in publications. Based in this, we included as much experimental information on EV
distribution as possible in our tables, including the distinction between the reporting of ‘EV detection’
in the brain (e.g. by fluorescent labels) and the ‘functional effects’ of EVs in the brain. It remains
however difficult to decide which read out can be considered as ‘solid data’ on brain targeting. While
a combination of these two read-outs would be the most solid data to prove EV brain targeting, a large
amount of the reported studies only provide one of these read-outs. Therefore, we compiled the
current research evidence, without stating strong conclusions and touched upon the issue in section
4.5 (revised manuscript line 915):
“Furthermore, distribution of EVs to specific regions does not necessarily imply functional delivery of
the EV cargo which will likely be a prerequisite for therapeutic applications. Specific strategies are
required to deduce functional delivery [6], but they are often not implemented to substantiate
conclusions in literature which adds a layer of complexity in judging whether a certain EV type has
therapeutic potential.”
To further highlight that in the future additional evidence is an important point that needs to be
addressed in these research fields, we provide the suggested deeper analysis of the experimental
details from table 1 and table 2, as described in the added Figure 3 (revised manuscript line 954,
rebuttal Figure 1), the text referring to this Figure (revised manuscript line 920) and in the conclusion
section (revised manuscript line 967):
Rebuttal Figure 1. Overview of the brain targeting EV detection methods of the summarized
literature. Pursuing the combination of both EV tracing and proof of functional cargo delivery will
provide stronger evidence of EV brain targeting capacities.
“To illustrate this, in Figure 3 we provide an overview of the detection methods reported for the EV
types summarized in Table 1 and Table 2. Indeed, while a vast amount of studies have combined
labelled EV/EV cargo tracing with proof of functional EV cargo delivery as solid proof of EV brain
delivery, a large amount of studies have based the targeting evidence on only one of these readout
approaches.”
Additionally, we included the following passage in the conclusion section (revised manuscript line
967):
“Moreover, by pursuing the combination of both EV tracing and proof of functional cargo delivery in
further research, stronger evidence of the applicability of different (un)modified EV types should be
gathered.”
Whether the brain barriers remain intact in the employed animal models is another difficult point to
assess. In general, the barrier capacity is not or incompletely studied in the summarized literature.
Moreover, since disrupted barrier phenotypes vary amongst disease models (e.g. brain tumors versus
different neurodegenerative diseases) and many different barrier functionalities can be affected with
variable degrees of impact on different transport mechanisms, the exact barrier states themselves
need to be further investigated. Consequently, this is still a controversial point of discussion outside
the scope of this review.
In this review, we focused on studies investigating systemic EV administration routes only. Since other
administration routes were not described, we decided not to include this sub-analysis. We added,
however, the following passage to highlight the influence of the administration route on the EV
potential in section 4.5 (revised manuscript line 927):
“Indeed, while the IV administration has been the most explored administration route over the past
decade [7], other systemic or local administration routes lead to different pharmacokinetics and
pharmacodynamics, which might be more or less suitable in specific disease contexts.”
4. EV storage is critical for their applications. While the authors spent 3 big paragraphs to identify
the issues with all conventional storage approaches, unexpectedly, they only briefly
mentioned the emerging effective EV preservation approaches in one sentence and referred
to one review in 2018. In my opinion, the authors are expected to spend more effort in
elaborating the emerging lyophilization approach to provide/propose the readers with
“solutions” instead of repeating “problems” that have been repeated many times in other
reviews. The authors are recommended to discuss two recent efforts (PMID: 27824088; PMID:
34371743) on EV/EV memetics lyophilization and the preservation of EV features both in vitro
and in vivo at 4C or room temperature for long term storage.
With our section on storage conditions, we primarily aimed to identify hurdles associated with the
conventional storage approaches as these approaches are currently used in research investigating the
brain targeting potential of (un)modified EVs. As such, we wanted to give the reader a critical view on
applied storage conditions and their influence on the interpretation and comparison of experimental
outcomes whilst using this opportunity to highlight the importance of reporting experimental
variables. An in-depth discussion of alternative preservation methods warrants a review on its own,
but in accordance with the reviewer’s comments we discussed the proposed papers and referred to
additional published reviews as follows (revised manuscript line 754):
“Several strategies to improve preservation of EVs are currently under investigation, including the
addition of a cryoprotectant such as trehalose. Storage of EVs in trehalose-containing PBS prevented
changes in EV number and size induced by consecutive freeze-thaw cycles and improved conservation
of biological activity of EVs upon storage at -80°C for one month [8]. Furthermore, trehalose can be
used as a so-called lyoprotectant during lyophilization (i.e. freeze-drying) which provides an alternative
to conventional EV storage in a buffer. Trehalose-lyoprotected, lyophilized EVs were shown to retain
their biological activity [9], identifying this preservation method as an interesting approach to be
explored further. For a more elaborate discussion on these alternative EV preservation methods, we
refer to reader to other reviews [10-12].”
5. In section 4.2, it is indeed wise to provide a brief summary of “EV isolation techniques” instead
of a much detailed discussion as this topic has been extensively reviewed elsewhere (ref 3, 95,
168, and a very recent comprehensive comparison on the pros & cons of current EV isolation
methods by PMID: 34229852). However, the authors may elaborate the discussion by
providing the audience certain guidance specifically for brain delivery based on the pros &
cons of available EV isolation approaches: what types of approaches may be better than the
others, in what circumstances?
In agreement with this comment, we adapted the manuscript sections as follows (revised manuscript
line 820):
“When aiming to use EVs as therapeutic moieties it will be key to balance two important goals: (1)
isolation of a pure EV sample, which will probably require the combination of different EV isolation
methods thereby enhancing processing time and implying loss of material in consecutive steps and (2)
applying an isolation method that allows processing large sample volumes in a time-efficient manner,
probably resulting in a less pure EV sample. Given the high number of EVs needed for brain targeting
studies, the second approach may be preferable although the importance of EV purity to maintain the
biological purpose will have to be determined. Similar to the EV storage conditions, currently no
consensus has been reached about a preferred EV isolation technique when assessing therapeutic EVs.
Nonetheless, we hypothesize that it is unlikely that a one-size-fits-all approach will become available
but instead, isolation techniques may need to be tailored to the EV source and/or the intended purpose.
Anyhow, an absolute prerequisite will be to retain the biological structure and function of EVs used for
brain delivery approaches. In this sense, harsh ultracentrifugation-based techniques or chemical
precipitation agents might be less suitable whereas gentle methods such as size exclusion
chromatography seem more favorable. However, comparative studies are warranted to confirm these
speculations.”
Minor points
6. English grammar checks are needed.
We carefully revised our manuscript for English grammar.
7. Section title of 3.3 should be rephrased to make it concise.
We rephrased the section title from “Extracellular vesicle brain barrier targeting and crossing” to
“Extracellular vesicle brain targeting” (revised manuscript line 302).
8. Subsection “3.3.2Brain (cancer) cell derived EVs” is not at the same level as 3.3.1, suggest to
include it as part of subsection 3.3.1.
We are not sure what is meant by this comment as “brain (cancer) cell derived EVs” is not a numbered
section in the manuscript. However, to enhance readability we labeled all subsections under section
3.3.1 and 3.3.2 as follows:
3.3.1 Unmodified extracellular vesicles
a. Brain (cancer) cell derived EVs
b. Cancer cell derived EVs
c. Stem cell derived EVs
d. Blood cell derived EVs
e. Other cellular sources
3.3.2 Modified extracellular vesicles
a. Rabies virus glycoprotein modified EVs
b. Other targeting strategies
References
1. Engelhardt, B.; Sorokin, L. The blood–brain and the blood–cerebrospinal fluid barriers:
function and dysfunction. Seminars in Immunopathology 2009, 31, 497-511,
doi:10.1007/s00281-009-0177-0 PMID - 19779720.
2. De Bock, M.; Vandenbroucke, R.E.; Decrock, E.; Culot, M.; Cecchelli, R.; Leybaert, L. A new angle
on blood-CNS interfaces: a role for connexins? FEBS Lett 2014, 588, 1259-1270,
doi:10.1016/j.febslet.2014.02.060.
3. Kadry, H.; Noorani, B.; Cucullo, L. A blood–brain barrier overview on structure, function,
impairment, and biomarkers of integrity. Fluids and Barriers of the CNS 2020, 17, 69,
doi:10.1186/s12987-020-00230-3 PMID - 33208141.
4. Redzic, Z. Molecular biology of the blood-brain and the blood-cerebrospinal fluid barriers:
similarities and differences. Fluids and Barriers of the CNS 2011, 8, 3, doi:10.1186/2045-8118-
8-3 PMID - 21349151.
5. Engelhardt, B.; Vajkoczy, P.; Weller, R.O. The movers and shapers in immune privilege of the
CNS. Nat Immunol 2017, 18, 123-131, doi:10.1038/ni.3666.
6. Verweij, F.J.; Balaj, L.; Boulanger, C.M.; Carter, D.R.F.; Compeer, E.B.; D’Angelo, G.; Andaloussi,
S.E.; Goetz, J.G.; Gross, J.C.; Hyenne, V., et al. The power of imaging to understand extracellular
vesicle biology in vivo. Nat Methods 2021, 18, 1013-1026, doi:10.1038/s41592-021-01206-3
PMID - 34446922.
7. Castilla, P.E.M.d.; Tong, L.; Huang, C.; Sofias, A.M.; Pastorin, G.; Chen, X.; Storm, G.; Schiffelers,
R.M.; Wang, J.-W. Extracellular vesicles as a drug delivery system:A systematic review of
preclinical studies. Adv Drug Deliver Rev 2021, 175, 113801, doi:10.1016/j.addr.2021.05.011
PMID - 34015418.
8. Bosch, S.; Beaurepaire, L.d.; Allard, M.; Mosser, M.; Heichette, C.; Chrétien, D.; Jegou, D.; Bach,
J.-M. Trehalose prevents aggregation of exosomes and cryodamage. Sci Rep-uk 2016, 6, 36162,
doi:10.1038/srep36162 PMID - 27824088.
9. Neupane, Y.R.; Huang, C.; Wang, X.; Chng, W.H.; Venkatesan, G.; Zharkova, O.; Wacker, M.G.;
Czarny, B.; Storm, G.; Wang, J.-W., et al. Lyophilization Preserves the Intrinsic Cardioprotective
Activity of Bioinspired Cell-Derived Nanovesicles. Pharmaceutics 2021, 13, 1052,
doi:10.3390/pharmaceutics13071052 PMID - 34371743.
10. Kusuma, G.D.; Barabadi, M.; Tan, J.L.; Morton, D.A.V.; Frith, J.E.; Lim, R. To Protect and to
Preserve: Novel Preservation Strategies for Extracellular Vesicles. Front Pharmacol 2018, 9,
1199, doi:10.3389/fphar.2018.01199 PMID - 30420804.
11. Yuan, F.; Li, Y.-M.; Wang, Z. Preserving extracellular vesicles for biomedical applications:
consideration of storage stability before and after isolation. Drug Delivery 2021, 28, 1501-
1509, doi:10.1080/10717544.2021.1951896 PMID - 34259095.
12. Bahr, M.M.; Amer, M.S.; Abo-El-Sooud, K.; Abdallah, A.N.; El-Tookhy, O.S. Preservation
techniques of stem cells extracellular vesicles: a gate for manufacturing of clinical grade
therapeutic extracellular vesicles and long-term clinical trials. Int J Vet Sci Medicine 2020, 8, 1-
8, doi:10.1080/23144599.2019.1704992 PMID - 32083116.

Round 2

Reviewer 2 Report

All my concerns are well addressed and the manuscript has been significantly improved. No further comments.